# Do manual therapies have a specific autonomic effect? An overview of systematic reviews

**Sonia Roura**[1]*, **Gerard Álvarez**[1,2], **Ivan Solà**[2], **Francesco Cerritelli**[3]

**1** Spain National Center, Foundation COME Collaboration, Barcelona, Spain, **2** Iberoamerican Cochrane Centre–Biomedical Research Institute Sant Pau, IIB Sant Pau, Barcelona, Spain, **3** Italian National Centre, Foundation COME Collaboration, Pescara, Italy

* soniaroura6@gmail.com

## Abstract

### Background

The impact of manual therapy interventions on the autonomic nervous system have been largely assessed, but with heterogeneous findings regarding the direction of these effects. We conducted an overview of systematic reviews to describe if there is a specific autonomic effect elicited by manual therapy interventions, its relation with the type of technique used and the body region where the intervention was applied.

### Methods

We conducted an overview according to a publicly registered protocol. We searched the Cochrane Database of Systematic Reviews, MEDLINE, EPISTEMONIKOS and SCOPUS, from their inception to march 2021. We included systematic reviews for which the primary aim of the intervention was to assess the autonomic effect elicited by a manual therapy intervention in either healthy or symptomatic individuals. Two authors independently applied the selection criteria, assessed risk of bias from the included reviews and extracted data. An established model of generalisation guided the data analysis and interpretation.

### Results

We included 12 reviews (5 rated as low risk of bias according the ROBIS tool). The findings showed that manual therapies may have an effect on both sympathetic and parasympathetic systems. However, the results from included reviews were inconsistent due to differences in their methodological rigour and how the effects were measured. The reviews with a lower risk of bias could not discriminate the effects depending on the body region to which the technique was applied.

### Conclusion

The magnitude of the specific autonomic effect elicited by manual therapies and its clinical relevance is uncertain. We point out some specific recommendations in order to improve the quality and relevance of future research in this field.

**Data Availability Statement:** All relevant data are within the manuscript and its Supporting Information files.

**Funding:** Funding for the publication of this article was provided by Registro de Osteopatas de España

(ROE). The funders had no role in study design, data collection and analysis, decision to publish, or preparation of the manuscript.

**Competing interests:** The authors have declared that no competing interests exist.

**Abbreviations:** ANS, Autonomic Nervous System; SNS, Sympathetic Nervous System; PNS, Parasympathetic Nervous System; MT, Manual Therapy; SR, Systematic Review; PROMs, Patient-Reported Outcomes; RCT, Randomized clinical trial; SC, Skin Conductance; ST, Skin Temperature; BP, Blood pressure; HR, Heart rate; HRV, heart rate variability; RR, Respiratory Rate; PC, Plasma Catecholamine; SBF, Skin Blood Flow; LF, Low Frequency; HF, High Frequency; RoB, Risk of bias.

## Introduction

The Autonomic Nervous System (ANS) is a system that works involuntarily, maintaining the body's internal environment (homeostasis) [1, 2]. It is classically differentiated between Sympathetic (SNS), Parasympathetic (PNS), and Enteric branches but from a physiological and neurochemical point of view has five components: the sympathetic noradrenergic system, the sympathetic cholinergic system, the parasympathetic cholinergic system, the sympathetic adrenergic system and the enteric nervous system. These components respond differently depending on the stressors and the pathophysiological states [3]. Several nuclei regulate ANS along the neuraxis, which reacts effectively to many internal (interoceptive) and external (exteroceptive) stimuli. Measuring the ANS within such complexity is challenging. Indeed, several tools and metrics have been developed and used to assess the ANS function [4–8]. "Table 1", adapted from Chiera et al. [9], summarises the autonomic markers' interpretation.

Manual therapy (MT) is defined as any touch-based conservative treatment approach that includes skilled hands-on techniques to assess and treat different symptoms and conditions using touch as exteroceptive solicitation [10]. It is used by a wide variety of professionals, including physical therapists, osteopaths, and chiropractors, and its use among different age groups and pathologies has been steadily increasing since 2000 [10]. MT includes a wide range of techniques such as soft tissue techniques, joint mobilisations or manipulations, massage, myofascial release, nerve manipulation, strain/counterstrain, and acupressure [11–13]. Of note, MT consists of applying these techniques and encompasses a person-centred approach based on a diagnostic clinical reasoning process, a conscientious patient/practitioner interaction and paying attention to patient re-education and advice [12]. The most common complaints treated by MT practitioners are low back and neck pain, sciatica, headache and temporomandibular disorders [14–18]. Recent systematic reviews suggest that the MT approach is clinically effective in treating chronic nonspecific neck pain, low back pain and pelvic girdle pain during pregnancy [19–21]. Moreover, several clinical studies have shown the effectiveness of MT in a wide variety of clinical conditions, including musculoskeletal pain among different age groups of patients [18, 22–27].

Various mechanisms of how MT affects neurobiology have been hypothesised by different researchers, suggesting that the manual solicitations applied by MT intervention produce neurophysiological responses able to modulate the pain experience [12, 28–30]. These responses can occur at three levels: 1) peripheral, that is, at the tissue level, where the application of MT induces a modulation of inflammatory response after tissue injury [31, 32]; 2) spinal: mechanical solicitations activate somato-autonomic reflexes, which in turn produce indirect neuromuscular responses and trigger intrinsic spinal networks through spino-spinal loops [33]; and 3) supraspinal, the use of manual contact might regulate brain areas like anterior cingulate cortex, amygdala or periaqueductal grey, which are crucial, for example, in pain experience, autonomic responses and hypoalgesia [28, 29]. Interestingly, these three levels were considered by King and colleagues as the key elements of the autonomic response of MT [34]. Indeed, the authors theorised that these levels are systematically involved during nociception, pain and inflammation [34]. It has been hypothesised that the effects induced by manual solicitations are dependent upon a specific type of touch, namely affective touch [35]. This gentle solicitation selectively activates low mechanical threshold C fibres [named C-tactile fibres or CTs] [35–39], which produce a specific activation of autonomic supraspinal nuclei as well as brain areas regulating emotions and interoception [36, 37, 39, 40]. Initial evidence suggests that MT interventions cause changes in different autonomic markers [41–49] and a recent paper has introduced the potential preventive role of MT in ANS imbalance [50]. However, the precise mechanisms by which MT interventions activate the autonomic response are still under exploration [50].

**Table 1. Autonomic markers and their interpretation.**

| Autonomic marker Tool | Specific metrics | Interpretation |
|---|---|---|
| **HRV** | **Time-domain** | |
| | SDNN | Standard deviation of NN intervals. It is highly correlated with ULF, VLF and LF. It is more accurate when calculated over 24h. An increase indicates a parasympathetic activation |
| | NN50 | Numbers of consecutive NN intervals that differ more than 50ms. An increase indicates a parasympathetic activation |
| | RMSSD | Root mean square of consecutive RR intervals. It is considered one of the main measures to assess vagal activity. It is similar to the non-linear metric SD1. An increase indicates a parasympathetic activation |
| | pNN50 | Percentage of NN50. It is correlated to RMSSD and HF power. An increase indicates a parasympathetic activation |
| | **Frequency-domain** | |
| | ULF (Power) | Ultra-low frequency value. Non-consensus regarding the mechanisms underlying ULF power. Very solo-acting biological processes, such as circadian rhythms, are implicated |
| | VLF (Power) | Very low-frequency value. Related to the heart's intrinsic nervous system, which generates VLF rhythm when afferent sensory cardiac neurons are stimulated. SNS activity due to physical and stress responses influences its oscillations' amplitude and frequency |
| | LF (Peak, Power, normalised units) | Low-frequency value. Non-specific index that reflects baroreceptor activity, it contains contributions of both the sympathetic and parasympathetic influences. High values of LF indicate a sympathetic predominance |
| | HF (Peak, Power, normalised units) | High-frequency value. Expression of parasympathetic activity, it corresponds to the HR variations related to the respiratory cycle known as RSA. It changes according to vagal modulation but does not reflect vagal tone. High values of HF indicate a parasympathetic predominance |
| | LF/HF ratio | Used to estimate SNS and PNS balance, although LF does not purely represent SNS, and PNS and SNS interact in a complex non-linear manner. A reduction of the LF/HF ratio indicates a sympathovagal balance |
| | **Non-linear** | |
| | SD1 | Poincare plot standard deviation. It correlates with baroreflex sensitivity, defined as the change in IBI duration per unit of change of BP, and HF. An increase indicates a parasympathetic activation |
| | SD2 | Poincare plot standard deviation. Correlates with LF power and baroreflex sensitivity. High values of LF indicate a sympathetic predominance |
| | SD1/SD2 | Ratio between SD1 and SD2, it measures the unpredictability of the RR time series. It correlates with the LF/HF ratio. Values over 1 indicates a parasympathetic effect, whereas values below 1 show a sympathetic effect |
| | DFA$\alpha$ | Detrended fluctuation analysis describing short-term ($\alpha 1$) or long-term ($\beta 2$) fluctuations. It is considered a sensitive parasympathetic index. *A decrease indicates a parasympathetic activation* |
| **Heart Rate** | Response to: breathing, Valsalva manoeuvre or postural change | The variability of heartbeat is used to assess cardiac sympathovagal function. An increase is considered a sympathetic effect, whereas a reduction shows parasympathetic activation |
| **Blood Pressure** | Response to: Valsalva manoeuvre, isometric exercise or postural change | Variation of blood pressure is utilized to assess adrenergic sympathetic function |
| **Microneurography** | | An electrophysiological technique used for recording single or multi-unit nerve traffic directly from human peripheral nerves. It permits to elucidate and quantify the sympathetic nerve activity in muscle and skin |
| **Spillover** | urine or plasma | A neurochemical technique employed to assess SNS based on the plasma or urine noradrenaline concentration |
| **Pupil Light Reflexes** | | A neurophysiological method assessing the dilation or restriction of the pupil. Parasympathetic action evokes pupil constriction whereas sympathetic noradrenergic activity produces pupil dilatation |
| **Electrodermal activity** | Galvanic skin response | A method utilized to measure neurally-mediated effects on sweat gland permeability—observed as changes in the resistance of the skin. It is considered a reliable measure of sympathetic cholinergic activation |
| **Thermal InfraRed Imaging** | | A method employed to measure the variability of temperature within specific areas of the face. Increase or decrease of the detected temperature implies, respectively, a parasympathetic or sympathetic activity |
| **Skin blood flow** | Response to: hand grip, cold, heat, baseline variation | Different methods are implied. The most common is laser Doppler. It is mainly used to evaluate SNS. A decrease is interpreted as an increase of sympathetic outflow |

It is well-known that a balanced ANS function is generally associated with health [50], while impairments in autonomic regulation have been considered a risk factor for physical and psychological morbidities and pathologies (e.g., hypertension, persistent generalising pain disorders, rheumatic diseases, diabetes or depression) [34, 51, 52]. It is worth noting that these clinical conditions are classified as dysautonomic, which means that they are generally associated with an increased sympathetic or decreased parasympathetic vagal activity. A recent systematic review (SR) investigated whether heart rate variability (HRV) parameters are altered in people with chronic low back pain when compared to healthy controls, showed that patients with chronic low back pain have a significant reduction in HRV, with sympathetic predominance compared to healthy controls [53].

Evidence about autonomic effects of manual therapy interventions is synthesised within many reviews [43, 45, 46, 49]. However, no clear conclusions have been yielded, and a comprehensive overview of systematic reviews in this area is currently lacking.

Considering the effects of MT on ANS function and the ANS role in pathologies, it is paramount to understand the potential effect of MT on ANS, which could be used as an adjunct therapy, even prevention, for diseases associated with autonomic imbalance.

This overview aims at describing if there is a specific autonomic effect elicited by MT interventions, its relation with the type of technique used and the body region where the MT was applied. The paper also explores how the effects reported are related to the measures used to assess the ANS and, eventually, the clinical relevance and applicability of the results. Finally, the review proposes some recommendations for future research in the field.

## Objectives

This overview aims to summarise the evidence published in SRs on the autonomic effects of MT interventions in either healthy or symptomatic populations. To this end, the proposed overview will answer the following questions:

1. Is there a specific autonomic effect observed after manual therapy interventions?

2. Different types of manual techniques elicit different autonomic effects?

3. Does the body region where the manual therapy is applied influence the autonomic outflow?

4. What are autonomic measures used to assess the autonomic effect elicited by manual therapy interventions?

5. Are the effects of MT on ANS clinically relevant?

## Materials and methods

The scope of this overview was to summarize systematic reviews with the primary aim of assessing the autonomic effect elicited by an MT intervention in either healthy or symptomatic populations, irrespective of their age.

According to a protocol registered at the Open Science Framework (DOI 10.17605/OSF. IO/TX69Y), we conducted an overview of systematic reviews that adhered standardised methodological guidance [54, 55]. As reporting standards for overviews are still under development [54] we report the findings of our study adhering the applicable items from the PRISMA statement [56].

## Inclusion criteria

We included systematic reviews that were defined according the following specific criteria [57].

i. provided specific eligibility criteria allowing to define the clinical question to be addressed in this overview [58]

ii. described a search strategy in at least two information sources [59]

iii. included a formal assessment of the risk of bias for the studies included in the review [58]

We defined the following criteria to answer the research question:

Eligible reviews had to include studies in healthy or symptomatic populations irrespective of their gender and age. Those studies had to assess the impact of any type of MT intervention isolated or in combination with additional interventions, compared to any type of control intervention (usual care, placebo or active interventions).

We defined MT as any touch-based intervention delivered with therapeutic intent. We used a standardised glossary to classify the MT techniques assessed in the eligible reviews [60]: mobilisations [passive movements that consist of oscillatory techniques and low frequency/ high amplitude techniques], manipulations (encompass spinal manipulative treatment, high-velocity low amplitude techniques, thrust techniques), myofascial techniques, balance ligamentous techniques, balance membranous techniques, cranial techniques, and soft tissue techniques (i.e. massage, passive stretching).

We included isolated interventions assessing an autonomic effect and combinations of techniques/approaches of manual interventions. We excluded packages of care where manual therapy was in combination with other therapies (i.e. exercises, cognitive education). We also excluded SRs assessing the effects of different interventions (not only MT) on an autonomic outcome. We limited eligibility to reviews focused on human studies and published in the English language.

The primary outcome for the overview was to assess the autonomic effect resulting from the MT intervention measured by the reaction of the SNS, PNS and /or the balance between SNS and PNS activity. For synthesis and analysis within the overview, the outcomes had to be measured using markers of autonomic response as described in "Table 1". Our secondary outcome of interest was to describe the clinical relevance of the results assessed by the outcomes related to pain improvement or any other patient-related outcomes (PROMS) reported in the included reviews.

## Search methods for identification of the reviews

We searched the Cochrane Database of Systematic Reviews (The Cochrane Library), MEDLINE (PubMed), EPISTEMONIKOS and SCOPUS from inception to March 2021. We designed search strategies tailored to the requirements of each database, combining terms from their controlled vocabulary (e.g. MeSH terms in MEDLINE) and text terms related to ANS and MT interventions. We report the MEDLINE search strategy included in "S1 Table".

Additionally, we tracked back and forwarded references and citations for the relevant studies through the Web of Science (Clarivate).

## Data collection and analysis

**Selection of reviews.** Two authors (SR and GA) independently assessed titles and abstracts of records identified by the electronic searches according to the inclusion criteria and decided on eligibility obtaining a full-text copy from relevant references.

We solved disagreements involving a third author (FC) to reach a consensus through discussion.

**Data extraction and assessment of the methodological quality of the included reviews.** Two overview authors (SR and GA) extracted data independently and discussed discrepancies until a consensus was reached. We planned in the protocol to involve a third author (FC) to solve disagreements. We used a data collection form specifically designed and piloted for the overview purposes ("S2 Table").

We extracted data on the following key features of each review:

a. identification elements (Title, author, year of publication, journal, number and type of articles included).

b. the characteristics of the patients included (healthy or symptomatic, age, gender), the generic and the name of the experimental intervention, the type of control used, number of sessions and duration of treatment and follow-up, outcome measures and the autonomic marker used.

c. Autonomic effect, duration of the effect (for this review, the effects were classified considering time as follows: short-term effect (immediately after the intervention), medium-term effect (from 1 to 24 weeks) and a long-term effect (more than 24 weeks) [61–63]), and hypothesis for effect found.

d. PROMS associated with the autonomic effect, contextual and confounding factors reported.

e. Design or reporting guidelines used and limitations and implications for future research if reported.

Using an Excel spreadsheet, we mapped the studies included within each included review to explore their overlap. We used the GROOVE tool [64] to assess the percentage of overlap among reviews. Percentage of the corrected covered area (CCA) was calculated considering: CCA <5% (slight overlap); CCA 5 - <10% (moderate overlap); CCA 10 - <15% (high overlap); CCA >15% (very high overlap).

We also used the GROOVE tool to decide which reviews should inform the overview findings and assess the degree of concordance among reviews conclusions.

Data from the primary studies of each SR (as documented in the published SRs) was extracted, including participants, intervention, comparison, outcome assessment, results and quality assessment.

Two authors (SR and GA) independently used the ROBIS tool [65] to appraise the methodological quality of the included reviews.

## Data analysis and synthesis

We synthesised in tables the included reviews characteristics and summarised findings narratively according to quality and outcomes of interest for the overview. We calculated the overlap between the included reviews according to the percentage of the corrected covered area [64].

Initially, we planned to perform a meta-analysis according to a two-step frequentist approach (random-effects model) with continuous end-point data for each outcome in R statistical software. We planned to assess within- and across-condition heterogeneity with the $I^2$ statistic, setting a threshold of $< 75\%$, to make the decision to pool the estimates across 1) the within condition reviews and 2) across the condition estimates. We planned to express effect estimates in standardised mean differences with 95% confidence intervals for all analyses. We

also planned to conduct Egger's test to detect publication bias in those analyses, including more than 10 studies. Finally, we anticipated in our protocol to explore subgroup effects according to the ages of participants (children and adolescents, adults, older adults), type of manual therapy intervention, comparator groups and duration of follow-up. The heterogeneity between studies did not allow us to perform pooled analysis.

## Generalisation of the evidence

We discussed on the generalisability of the overview findings and the applicability of the body of primary research assessed through the included reviews. We analysed and interpreted the data obtained from the included reviews and classified them around a list of predefined questions which we anticipated to be relevant:

1. Is there evidence of a general autonomic effect of manual therapy interventions across reviews?

2. Had this effect a specific ANS direction (i.e. sympathetic, parasympathetic)?

3. Is there evidence of the relation between the autonomic effect and the decrease of pain or any other clinical outcome?

4. Is this effect different when using an isolated MT technique or combining different MT techniques/approaches?

5. Is the effect robust across conditions, type of interventions and age groups?

6. Is the body region where the technique is applied important to observe any specific effect?

7. Is there evidence for the duration of the autonomic effect? (short, medium or long-term effect)

8. Do the contextual factors and/or the non-specific effects of MT interventions influence the autonomic effects?

9. Can we infer that the effect might be observed across conditions not included in the current overview?

To respond to these questions we considered the findings of both the included reviews and the assessed primary studies, their methodological quality, their conclusions and the limitations and implications for future research highlighted within the reviews.

## Results

### Search results and study eligibility

The search yielded 557 records (383 from MEDLINE, 1 from the CDSR, 35 from EPISTEMO-NIKOS and 138 from SCOPUS), from which we selected 17 reviews for a detailed assessment. We excluded 5 reviews for the following reasons: one for the research design [66], two that did not include a quality assessment of the primary studies [42, 67], one did not fulfil the outcome established for this overview [68], and one included a combination of different types of interventions [69]. As a result of the eligibility process, we included 12 systematic reviews in the overview. We display the complete process in a flowchart (Fig 1).

### Characteristics of included reviews

Characteristics of included reviews are shown in "Table 2" and reported as follows:

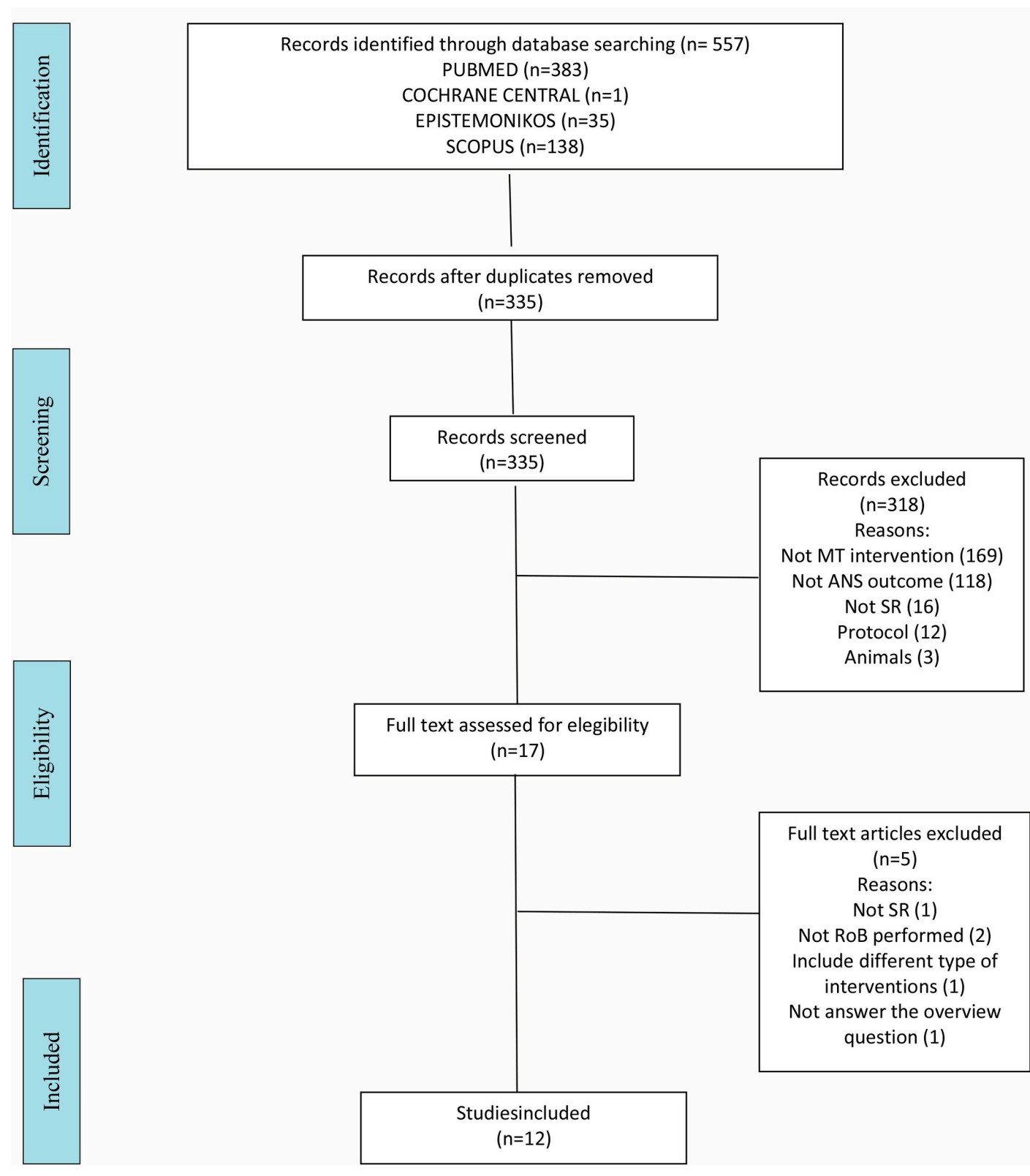

**Fig 1. PRISMA flowchart.**

**Table 2. Characteristics of the included reviews.**

| Included review | DESIGN | POPULATION | INTERVENTION | FINDINGS |
|---|---|---|---|---|
| **Wirth 2019** | **Risk of bias (ROBIS)** low **Search date** July 2018 **Study design** controlled studies **Included studies** 18 **Quality / bias approach** score (Downs and Black) **Meta-analysis** no **Assessed quality of evidence** no | healthy volunteers, patients with pain in different locations and of different chronicity | **Intervention** HVLA-SMT **Comparison** inactive control or placebo | **Measure** HRV, BP, SC, O2 saturation, pupillometry **Effect** changes in ANS, changes in heart rate variability and skin conductance, and the direction of effects depending on body region |
| **Picchiotino 2019** | **Risk of bias (ROBIS)** low **Search date** July 2018 **Study design** sham controlled trials **Included studies** 29 **Quality / bias approach** domains (Cochrane RoB tool) **Meta-analysis** yes **Assessed quality of evidence** yes | healthy volunteers and symptomatic patients | **Intervention** JMT applied on spinal or peripheral joints **Comparison** sham techniques | **Measure** SC, ST, SBF, HR, HRV, RR, PB, PC, pupilometry **Effect** sympathetic effect on the skin for mobilizations regardless the area treated, not in other autonomic markers |
| **Rechberger 2019** | **Risk of bias (ROBIS)** unclear **Search date** April 2018 **Study design** any design **Included studies** 23 **Quality / bias approach** score (Downs and Black, Kienle) **Meta-analysis** no **Assessed quality of evidence** no | healthy volunteers, craniofascial pain, temporomandibular dysfunction, hypertensive, lumbar pain, neck pain, headache | **Intervention** HVLA techniques, mobilizations, functional and cranial **Comparison** inactive control or placebo | **Measure** PC, BP, HR, SBF **Effect** changes in ANS |
| **Navarro-Santana 2019** | **Risk of bias (ROBIS)** unclear **Search date** March 2018 **Study design** RCT **Included studies** 18 **Quality / bias approach** domains (Cochrane RoB tool) **Meta-analysis** yes **Assessed quality of evidence** yes | healthy volunteers, elbow pain, neck pain, craniofacial pain | **Intervention** mobilization techniques **Comparison** placebo, sham or no intervention | **Measure** SC, ST **Effect** sympathetic-excitatory effect |
| **Araujo 2018** | **Risk of bias (ROBIS)** low **Search date** March 2014 **Study design** RCT **Included studies** 18 **Quality / bias approach** score (PEDro score) **Meta-analysis** no **Assessed quality of evidence** yes | healthy volunteers and symptomatic patients | **Intervention** spinal mobilizations and manipulations **Comparison** inactive control, placebo or other SMT techniques | **Measure** SC, ST, HR, pupilometry **Effect** uncertain |

*(Continued)*

**Table 2.** (Continued)

| Included review | DESIGN | POPULATION | INTERVENTION | FINDINGS |
|---|---|---|---|---|
| **Amoroso Borges 2018** | **Risk of bias (ROBIS)** unclear **Search date** December 2016 **Study design** RCT **Included studies** 10 **Quality / bias approach** score (PEDro score) **Meta-analysis** no **Assessed quality of evidence** | healthy volunteers | **Intervention** spinal manipulation and myofascial techniques **Comparison** inactive control or placebo | **Measure** HRV **Effect** changes in ANS |
| **Galindez 2017** | **Risk of bias (ROBIS)** unclear **Search date** August 2016 **Study design** RCT **Included studies** 11 **Quality / bias approach** score (Cochrane Back Review tool) **Meta-analysis** no **Assessed quality of evidence** no | healthy volunteers and symptomatic patients | **Intervention** cervical HVLA manipulation techniques **Comparison** any placebo, sham techniques, manual contact, quite rest | **Measure** ECG: HR, BP, oxygen saturation **Effect** a decrease in diastolic BP was found; heart rate, systolic BP, electrocardiogram, and bilateral pulse oximetry, the changes were not significant |
| **Lascurain 2016** | **Risk of bias (ROBIS)** high **Search date** not reported **Study design** RCT, controlled trials and case control studies **Included studies** 24 **Quality / bias approach** score (Cochrane Back Review tool) **Meta-analysis** no **Assessed quality of evidence** no | symptomatic patients (neck pain, lumbar pain, cervicobrachial neurogenic pain, shoulder, elbow, craniofacial pain) | **Intervention** spinal manipulation **Comparison** placebo, inactive control or no intervention | **Measure** SC, RR, HR, ST **Effect** increase SC, HR, RR, decrease or no change in ST |
| **Chu 2014** | **Risk of bias (ROBIS)** low **Search date** not reported **Study design** RCT **Included studies** 11 **Quality / bias approach** score (PEDro score) **Meta-analysis** yes **Assessed quality of evidence** no | healthy volunteers and chronic lateral epicondylalgia,cervico-craniofacial pain and nonacute cervicobrachial neurogenic pain. | **Intervention** SMT to the cervical or thoracic spine segments **Comparison** not described | **Measure** SC, ST, pain, ROM **Effect** sympathetic-excitatory effect |

*(Continued)*

**Table 2.** (Continued)

| Included review | DESIGN | POPULATION | INTERVENTION | FINDINGS |
|---|---|---|---|---|
| Kinsgton 2014 | **Risk of bias (ROBIS)** unclear<br>**Search date** May 2012<br>**Study design** RCT<br>**Included studies** 7<br>**Quality / bias approach** mixed (Cochrane RoB tool / PEDro score)<br>**Meta-analysis** no<br>**Assessed quality of evidence** no | healthy volunteers and symptomatic patients | **Intervention** spinal mobilization<br>**Comparison** placebo, inactive control or no intervention | **Measure** SC, ST, HR, RR, BP<br>**Effect** sympathetic-excitatory effect |
| Hegedus 2011 | **Risk of bias (ROBIS)** unclear<br>**Search date** November 2010<br>**Study design** controlled trials<br>**Included studies** 10<br>**Quality / bias approach** score (Cochrane Back Review tool)<br>**Meta-analysis** no<br>**Assessed quality of evidence** yes | healthy volunteers and symptomatic patients | **Intervention** spinal mobilization<br>**Comparison** not described | **Measure** SC, ST<br>**Effect** sympathetic-excitatory effect |
| Schmid 2008 | **Risk of bias (ROBIS)** unclear<br>**Search date** November 2007<br>**Study design** RCT<br>**Included studies** 15<br>**Quality / bias approach** score (Cochrane Back Review tool)<br>**Meta-analysis** no<br>**Assessed quality of evidence** no | healthy volunteers and symptomatic patients (neck or upper limb) | **Intervention** passive accessory cervical joint mobilisation techniques<br>**Comparison** one or two control conditions, receiving either manual contact, no contact or therapeutic ultrasound interventions | **Measure** SC, BP, HR, RR<br>**Effect** sympathetic nervous system excitation (increase SC, no effect on ST) |

RCTs: randomized clinical trials; CTs: clinical trials; HVLA: high velocity low amplitude techniques: SMT: spinal manipulative therapy; JMT: joint manipulative therapy; HRV: heart rate variability; SC: skin conductance; ST: skin temperature; HR: heart rate; RR: respiratory rate; BP: blood pressure; ROM: range of movement; ECG: electrocardiogram; SBF: skin blood flow; ANS: autonomic nervous system.

## Types of studies

All the reviews included randomized clinical trials (RCTs), one also included quasi-RCTs, four included non-randomized trials and two other designs.

There were 194 primary studies included in the 12 reviews. Considering duplicates, a total of 101 primary articles were analyzed. Because 39 of these primary studies did not assess the ANS, the analysis was conducted using 62 studies (enrolling 2201 participants).

## Participants

Ten out of 12 reviews included both genders and adult healthy and symptomatic participants, whereas two included only adult symptomatic participants [43, 70]. Specific conditions varied

**Table 3. Body regions where the studies were focused.**

| Body region | Systematic Reviews |
|---|---|
| Spine (all regions) | Hegedus 2011, Kingston 2014, Amoroso 2018, Araujo 2018 Navarro 2019, Picchiottino 2019, Wirth 2019 |
| Only cervical spinal | Schmid 2008, Lascurain-Aguirrebeña 2016, Galindez-Ibarbengoetxea 2017 |
| Cervical and thoracic spine | Chu 2014 |
| Peripheral joints | Navarro-Santana 2019, Picchiottino 2019 |
| All body regions | Rechberger 2019 |

among reviews and included: lateral epicondylalgia, cervical pain, cervicobrachial pain, craniofacial pain, lumbar pain, shoulder pain, elbow pain and hypertensive subjects.

## Interventions

Nine out of 12 reviews included spinal mobilizations [43–45, 47–49, 71–73], of which only one focused on the cervical spine [72]. Seven reviews included spinal manipulations [41, 45, 46, 48, 49, 70, 72]. Three included cranial techniques [44, 48, 71], two myofascial techniques [44, 48] and one peripheral mobilizations [46]. A list of the body regions where the studies were focused can be found in "Table 3".

Most of the reviews assessed the autonomic effect of a single isolated technique. One of the 12 reviews included studies combining different techniques as a therapeutic approach (manipulations, mobilizations, myofascial and cranial techniques) [48].

## Comparisons

Included reviews explored comparisons of interventions with no treatment, placebo/sham or other interventions. Seventy-five per cent of the included studies used a placebo/sham comparator. Comparison groups included within each review are summarized in "Table 2".

## Outcomes

Included reviews covered a wide range of outcomes. Three reviews assessed the autonomic skin activity only [43, 46, 73]. One assessed only cardiovascular autonomic activity [44]. The rest combined different assessments, including skin and cardiovascular activity, pupil autonomic control and sympatho-adrenal activity.

The autonomic markers included in the reviews were: Skin conductance -SC- was the most frequent marker used (9/12; 75%), followed by skin temperature -ST- (8/12; 67%), heart rate -HR- (7/12; 59%), blood pressure -BP- (6/12; 50%), heart rate variability -HRV- (4/12; 33%), respiratory rate -RR- (4/12; 33%), pupillometry (4/12; 33%), oxygen saturation (3/12; 25%), plasma catecholamine -PC- (2/12; 17%) and skin blow flow -SBF- (1/12; 8%). When exploring the primary studies included in the reviews, the most frequent autonomic markers were SC (32%) and HRV (31%).

These markers were used to measure skin autonomic activity (SC, ST, SBF), cardiovascular autonomic activity (HRV, BP, HR, oxygen saturation), pupil autonomic regulation and sympatho-adrenal system (PC). Cardiovascular autonomic regulation was the most frequently assessed (48% of the primary studies), followed by skin autonomic activity (37%).

Seven of the 12 reviews included multiple assessments of the ANS. Moreover, 5% of the primary studies assessed also PROMS with pain measurements.

Table 4. Risk of bias of included reviews (ROBIS assessments).

| REVIEW | 1. Concerns regarding specification of STUDY ELEGIBILITY CRITERIA | 2. Concerns regarding methods used for the IDENTIFICATION AND SELECTION OF STUDIES | 3. Concerns regarding methods used in DATA COLLECTION AND STUDY APPRAISAL | 4. Concerns regarding methods used in SYNTHESIS AND FINDINGS | RISK OF BIAS IN THE REVIEW |
|---|---|---|---|---|---|
| Wirth 2019 | low | low | low | low | LOW |
| Picchiotino 2019 | low | low | low | low | LOW |
| Araujo 2018 | low | low | low | low | LOW |
| Chu 2014 | low | low | low | low | LOW |
| Navarro-Santana 2019 | unclear | low | low | unclear | UNCLEAR |
| Hegedus 2011 | unclear | low | low | unclear | UNCLEAR |
| Rechberger 2019 | low | unclear | unclear | unclear | UNCLEAR |
| Galindez 2017 | low | unclear | unclear | unclear | UNCLEAR |
| Kinsgton 2014 | low | unclear | low | high | UNCLEAR |
| Amoroso Borges 2018 | low | unclear | unclear | high | UNCLEAR |
| Schimid 2008 | low | low | low | high | UNCLEAR |
| Lascurain 2016 | unclear | unclear | low | high | HIGH |

## Risk of bias of included reviews

Five of the included reviews were assessed as low risk of bias (RoB) using the ROBIS tool, six scored unclear RoB and one high RoB, as summarised in "Table 4". Details of this assessment can be found in "S3 Table".

Four SRs reported a GRADE assessment of the certainty of the evidence of their included studies [45, 46, 49, 73]. In general, low to very low was the most rated quality of evidence for the autonomic effects of MT interventions and moderate evidence for mobilizations.

## Overlap between included reviews

"Fig 2" displays the overlap between the included reviews through the percentage of the corrected covered area and the studies overlapping in the included reviews. We observed a high overlap between the reviews that ranged from 10% to more than 15% CCA. We analyzed in detail this overlap in the discussion.

## Findings from the included reviews related to the autonomic effects of interventions, techniques used and the autonomic markers

The findings and conclusions of the included reviews are summarized in "Table 5". For those reviews that used GRADE we also show their judgements on the certainty of evidence. The concordance between the findings of the reviews with a high degree of overlap can be seen in "S4 Table". Eleven of the twelve reviews specifically addressed acute autonomic effects. Skin conductance was the only autonomic marker that demonstrated a consistent acute sympatico-excitatory effect, but only for spinal mobilisations, not for other techniques.

**Autonomic effects and techniques.** MT techniques eliciting PNS or SNS changes were different, as shown in "Table 5".

Ten out of 12 SRs concluded that mobilisations and manipulations have a sympathetic-excitatory effect on sympathetic skin activity [41, 43, 45–49, 70, 72, 73]. Confirmatory results

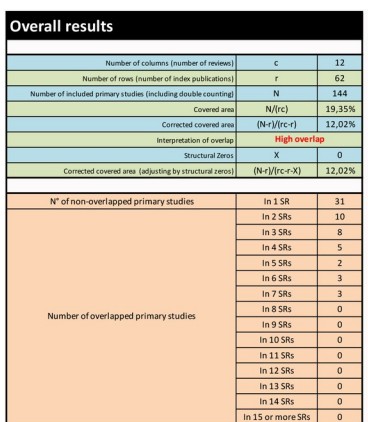

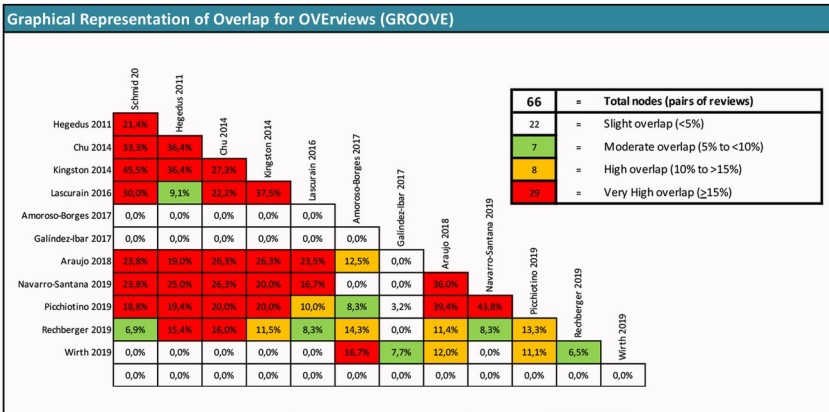

**Fig 2. Overlapping ANS outcomes.** Overlapping of the included reviews.

were also shown using other autonomic markers [41]. One study concluded that mobilisations have an SNS effect at the skin level but not when assessing other systems, such as cardiovascular and respiratory [45]. One SR found a PNS effect on the cardiovascular system after different MT interventions [44, 47]. Two SRs differentiated the effect of MT interventions (sympathetic or parasympathetic) depending on the area of the spine where the intervention was performed [44, 47]. One SR claimed that MT interventions elicit changes on ANS [48]. However, the authors did not mention either the direction of changes or any effect associated with body regions [48].

**Autonomic effects and autonomic markers.** *Assessing ANS with skin autonomic markers.* Ten SRs included autonomic skin markers to assess ANS activity [41, 43, 45–49, 70, 72, 73]. SC was used in all the articles and ST on eight SRs. Skin blood flow assessed with laser doppler flowmetry was used in two SRs ("Table 6").

All articles assessing SC showed an increase suggesting an SNS effect after manipulation, mobilization and myofascial techniques [41, 43, 45–47, 49, 70, 72, 73]. When assessing ST, most authors found a reduction after mobilisations in different body regions [41, 43, 46, 49]. Two reviews did not find any effect of mobilisations on ST [45, 70], and one found conflicting evidence [72]. Conversely, two SRs assessed autonomic skin activity by SBF. One showed an increase in blood flow when applying cervical mobilisations [72], and the other found an increase in blood flow when using peripheral mobilisations but no effect after spinal mobilisations [45].

*Assessing ANS with HRV.* Four SRs used HRV to assess ANS activity [44, 45, 47, 48]. Although one SR did not specify which markers were used to assess HRV, three SRs reported results on different HRV domains and metrics ("Table 7")—for a more comprehensive description of these metrics see [8, 9]-. In general, all the studies reported short-term measurements of HRV. Any of the reviews included linear and non-linear measurements.

When considering the time domain, an increase of all the indices (SDNN, RMSSD; pNN50) was found, indicating a PNS effect, independent of the techniques used and the body region where those techniques were applied. For the frequency domain, an increase in LF (without specifying if absolute or normalized values) was shown when applying manipulations to lower cervical and upper thoracic, suggesting an increased SNS activity [44, 47]. HF showed contradictory results depending on the technique and the body regions where the intervention was administered. A decrease in the HF absolute value was found when applying manipulations to the lower neck and upper thoracic [47]. An increase in HF absolute values was

**Table 5. Techniques, autonomic markers used and autonomic effects of included reviews.**

| Review | Technique | Measure | Effect | GRADE Assessment |
|---|---|---|---|---|
| **Wirth 2019** | | | | |
| | HVLA upper cervical | HRV | Increase PNS activity | NO |
| | | BP | reduction systolic BP, no effect | |
| | HVLA lower cervical | SC | Increase SNS activity | |
| | | HRV | Increase SNS activity | |
| | | PB | reduction systolic BP, no effect | |
| | HVLA upper thoracic | HRV | Increase SNS activity | |
| | | Oxygen saturation | no effect | |
| | | Pupillometry | no effect | |
| | HVLA lumbar | SC | Increase SNS activity (healthy)/ increase PNS activity (LBP) | |
| **General conclusion:** *changes in ANS associated with HVLA-SMT in pain patients, but the direction of change is not consistent across studies.* **Recommendations:** *baseline values of outcome measures and to link neurophysiological HVLA-SMT effects to changes in perceived pain* | | | | |
| **Picchiottino 2019** | | | | |
| | Mobilisations (oscillatory) | SC | Increase SNS activity (moderate evidence) | Moderate-certainty evidence (SC) |
| | | ST | no effect (no good markers for SNS skin activity) | |
| | | SBF | modulate or no effect | |
| | | HR | no acute effect / increase systolic BP | |
| | | HRV | no acute effect | |
| | | RR | increase RR (low evidence) | |
| | Mobilisations (movement) | SC/ST (spinal) | no acute effect | |
| | | SC (peripheric) | increase SNS activity (low evidence) | |
| | | ST/SBF | modulate SNS activity (very low evidence) | |
| | | HR/BP | modulate SNS activity (very low evidence) | |
| | HVLA | HRV | no acute effect | |
| | | HR | no acute effect | |
| | | BP | no acute effect | |
| | | pupil diameter | no acute effect | |
| | | PC | no acute effect | |
| **General conclusion:** *Contradictory, Uncertain and inconclusive because of low evidence and limited clinical relevance. Moderate evidence for sympathetic-excitatory effect after mobilizations regardless of the body region* | | | | |
| **General recommendation:** *Increase methodological quality, long-term effects and measures with pain, assess in chronic pain* | | | | |
| **Araujo 2019** | | | | |
| | Mobilisations | SC | increase, increase SNS activity | Very low-low evidence |
| | | ST chronic pain | decrease, increase SNS activity | |
| | | HR | increase, increase SNS activity | |
| | Manipulations | pupil diameter | no effect (vs placebo)/ decrease diameter (vs no treatment) | |
| **General conclusion:** *Conflicting results, no clinical improvement* | | | | |
| **Recommendations:** *Increase quality and reporting, long-term follow up* | | | | |
| **Chu 2014** | | | | |
| | Spinal MT on cervical pain | SC | increase, increase SNS activity | NO |
| | | ST | decrease, increase SNS activity | |
| | Spinal MT on lumbar pain | SC | increase, increase SNS activity | |
| | | ST | decrease, increase SNS activity | |
| **General conclusion:** *Sympatico excitatory but challenges for the current understanding of skin nervous activity (local endothelial mechanisms)* | | | | |

*(Continued)*

**Table 5.** (Continued)

| Review | Technique | Measure | Effect | GRADE Assessment |
|---|---|---|---|---|
| Recommendations: *Palpatory findings + pain scale, recommendations for using laser Doppler Flowmetry* | | | | |
| **Navarro-Santana 2019** | | | | |
| | Mobilisations | SC | increase, increase SNS activity | Very low-moderate evidence |
| | | ST | decrease, increase SNS activity | |
| General conclusion: *Heterogeneity, Results suggests a sympathetic effect (SC, ST) after mobilization. The authors used GRADE and rated the results as moderate certainty (or quality) of evidence.* | | | | |
| Recommendations: *Use of different manual therapy approaches, Association with treatment benefits* | | | | |
| **Hegedus 2011** | | | | |
| | Spinal mobilisations | ST | No change | Very low-low evidence |
| | | SC | increase, increase SNS activity | |
| General conclusion: *Sympatico excitatory effect* | | | | |
| Recommendations: *Randomized controlled trials on subjects with pain and decreased function* | | | | |
| **Rechberger 2019** | | | | |
| | HVLA cervical | HRV | Significant change, no conclusion | NO |
| | HVLA lumbar | HRV | Increase PNS activity | |
| | Cranial | HRV | Increase PNS activity | |
| | | PC +BP +HR | No effect | |
| | | SBF | slight reduction SNS activity | |
| | Mobilization thoracic | pain assessment | reduction in pain because of SNS changes | |
| | | SC | Increase SNS activity | |
| | | HRV | change in ANS | |
| General conclusion: *Changes in ANS but the direction of change is not consistent across studies. Inconclusive results* | | | | |
| Recommendations: *Increase the number of participants in studies. Increase methodological quality* | | | | |
| **Galindez 2017** | | | | |
| | Cervical HVLA | BP | Decrease in Systolic BP in hypertensive I, no effect in healthy subjects | NO |
| | | HR | No effect | |
| | | Oxygen saturation | No effect | |
| General conclusions: *A decrease in diastolic BP was found; however, for other studied variables, such as heart rate, systolic BP, electrocardiogram, and bilateral pulse oximetry, the changes were not significant* | | | | |
| Recommendations: *not reported* | | | | |
| **Kingston 2014** | | | | |
| | Spinal mobilisations | ST | decrease, increase SNS activity | NO |
| | | SC | increase, increase SNS activity | |
| | | RR | increase, increase SNS activity | |
| | | HR | increase, increase SNS activity | |
| | | BP | increase, increase SNS activity | |
| General conclusion: *Sympatico excitatory effect* | | | | |
| Recommendations: *symptomatic population* | | | | |
| **Amoroso-Borges 2018** | | | | |
| | Manipulation cervical and lumbar | HRV | increase PNS activity | NO |
| | Manipulation thoracic | HRV | increase PNS activity | |
| | Myofascial | HRV | increase PNS activity | |
| | Cranial | HRV | increase PNS activity | |

(*Continued*)

**Table 5.** (Continued)

| Review | Technique | Measure | Effect | GRADE Assessment |
|---|---|---|---|---|
| *General conclusion:* PNS response when stimulation was performed in the cervical and lumbar regions, SNS response when stimulation was performed in the thoracic region | | | | |
| *Recommendations:* Long-term effects | | | | |
| **Schmid 2008** | | | | |
| | Cervical mobilizations | SC | increase, increase SNS activity | NO |
| | | SBF | Increase in the elbow, decrease in the hand, increase SNS activity | |
| | | HR | increase, increase SNS activity | |
| | | BP | Increase diastolic BP, increase SNS activity | |
| | | ST | Conflicting evidence | |
| *General conclusions:* Sympatico excitatory effect after cervical mobilisation regardless of the body segment receiving the treatment. | | | | |
| *Recommendations:* To include outcome measures designed to evaluate the multisystem effects of treatment. Better reporting in order to facilitate future meta-analysis and comparison of results. | | | | |
| **Lascurain 2016** | | | | |
| | Mobilizations | SC | increase, increase SNS activity | NO |
| | | HR | increase, increase SNS activity | |
| | | RR | increase, increase SNS activity | |
| | | ST | No effect | |
| | | SBF | decrease, increase SNS activity | |
| *General conclusion:* Sympathoexcitation after mobilizations | | | | |
| *Recommendations:* To observe possible mechanisms of action and symptom reduction in patients. | | | | |

SNS: sympathetic nervous system; PNS: parasympathetic nervous system; HVLA: high-velocity low amplitude; HRV: heart rate variability; PC: plasma catecholamine; SC: skin conductance; ST: skin temperature; SBF: skin blow flow; RR: respiratory rate; HR: heart rate; BP: blood pressure; MT: manual therapy; SMT: spinal manipulative therapy; LBP: low back pain.

observed when manipulation was used to the upper neck [44] and yet an increase of the HF—both absolute values and normalized units—after myofascial techniques. The majority of the included papers assessing the LF/HF ratio found a decrease in this ratio, meaning a parasympathetic effect. This PNS effect was observed for manipulation techniques to the upper neck and lumbar spine and for myofascial techniques [44, 47]. Notwithstanding this PNS response, the LF/HF ratio was found to increase when applying manipulation techniques to the lower neck and upper thoracic spine, indicating an increased SNS response [44, 47]. One SR did not find any effect on HRV values (HF spectral power and LF/HF values) after mobilisations or manipulations [45].

*Assessing ANS with other autonomic markers.* Eight papers in our review used BP and HR. Three SRs found an increase in systolic blood pressure after manipulation of the neck [47] and after mobilisations [41, 45] and in the diastolic blood pressure after neck manipulations [72]. One SR found a decrease in systolic BP in hypertensive type I subjects but not in healthy participants [71]. Two studies found an increase in HR after mobilisations [49, 67], and one found no effect after cervical manipulation [71]. No effect was also found on BP and HR after either cranial techniques [48] or spinal manipulations [45].

As far as respiratory rate is considered, 3 systematic reviews took it into account as an autonomic marker. One SR argued that RR increases after oscillatory mobilisations even if the overall evidence was very low [45], confirming previous SRs [41, 70].

Concerning oxygen saturation, two articles did not show any effect after manipulations in the neck or upper thoracic area [47, 71].

**Table 6. Skin autonomic markers among the included studies.**

| Skin marker | Technique | Result | autonomic effect | Article |
|---|---|---|---|---|
| SC | HVLA lumbar | increase | SNS effect | Wirth 2019 |
| | mobilisations (oscillatory) | increase | SNS effect | Picchiottino 2019 |
| | spinal mobilizations (movement) | increase | SNS effect | Picchiottino 2019, Navarro-Santana 2019, Araujo 2019, Kingston 2014, Lascurain 2016, Schmid 2008, Hegedus 2011 |
| | peripheral mobilisation | increase | SNS effect | Picchiottino2019, Navarro-Santana 2019. |
| | spinal manual therapy | increase | SNS effect | Chu 2014 |
| ST | mobilisations (oscillatory) | no effect | no effect | Picchiottino 2019 |
| | spinal mobilisations (movement) | no effect | no effect | Picchiottino 2019, Hegedus 2011, Lascurain 2016 |
| | | decrease | SNS effect | Navarro-Santana 2019, Araujo 2019, Kingston 2014 |
| | | Conflicting evidence | | Schmid 2008 |
| | peripheral mobilisation | decrease | SNS effect | Picchiottino 2019, Navarro-Santana 2019 |
| | spinal manual therapy | decrease | SNS effect | Chu 2014 |
| Skin blood flow (LDF) | mobilisations (oscillatory) | no effect | no effect | Picchiottino 2019 |
| | peripheral mobilisation | increase | SNS effect | Picchiottino 2019 |
| | Cervical mobilizations | Increase in the elbow | SNS effect | Schmid 2008 |
| | | decrease in the hand | PNS effect | |

SC: skin conductance; ST: skin temperature; SBF: skin blood flow; HVLA: high velocity/low amplitude; SNS: sympathetic; PNS: parasympathetic; LDF: laser doppler flowmetry.

**Table 7. Short-term effects of different techniques on specific autonomic indices of HRV.**

| HRV marker | Technique | Result | autonomic effect | Article |
|---|---|---|---|---|
| *Time-domain* | | | | |
| SDNN | Myofascial | increase | PNS activation | Amoroso-Borges 2018 |
| RMSSD | Myofascial | increase | PNS activation | Amoroso-Borges 2018 |
| pNN50 | Myofascial | increase | PNS activation | Amoroso-Borges 2018 |
| *Frequency domain* | | | | |
| LF | HVLA upper thoracic | increase | SNS predominance | Wirth 2019 |
| | HVLA lower cervical | increase | SNS predominance | Wirth 2019 |
| HF | HVLA upper thoracic | decrease | Decreased PNS predominance | Wirth 2019 |
| | HVLA lower cervical | decrease | Decreased PNS predominance | Wirth 2019 |
| | HVLA upper cervical | increase | PNS predominance | Amoroso-Borges 2018 |
| | Myofascial | increase | PNS predominance | Amoroso-Borges 2018 |
| | Spinal manipulation | No effect | No effect | Picchiottino 2019 |
| LF/HF ratio | HVLA upper thoracic | increase | increased sympa- tho-vagal balance | Wirth 2019, Amoroso-Borges 2018 |
| | HVLA lower cervical | increase | increased sympatho-vagal balance | Wirth 2019 |
| | HVLA upper cervical | decrease | sympatho-vagal balance | Wirth 2019, Amoroso-Borges 2018 |
| | HVLA lumbar | decrease | sympatho-vagal balance | Amoroso-Borges 2018 |
| | Myofascial | decrease | sympatho-vagal balance | Amoroso-Borges 2018 |
| | Spinal Manipulations | No effect | No effect | Picchiottino 2019 |

SDNN: the standard deviation of all R–R intervals; RMSSD: the root mean square of successive differences; pNN50: the percentage of successive normal sinus RR intervals more than 50 ms; PNS: Parasympathetic nervous system; SNS: Sympathetic nervous system; HVLA: high-velocity low amplitude technique; LF: low frequency; HF: high frequency.

Three SRs assessed ANS with pupillometry. All of them found no changes in pupil diameter after manipulations compared to placebo [45, 47, 49]. However, one SR found that the pupil diameter decreased when comparing manipulations with no treatment instead of placebo [49].

Plasma catecholamine did not show any effect after manipulations and cranial techniques [45, 48].

## Quality assessment tools used in the included reviews

The reviews differed in the approach used to assess the internal validity of the primary studies which they included ("Table 8 and S5 Table"). Only three performed a domain- based assessment [41, 45, 46] and the rest used a variety of scores. For that reason, judgements on bias or quality of overlapping primary studies among the included SRs varied in some cases. Two reviews at low risk of bias that used the Downs and Black scale agreed in the score provided to the two studies included in both reviews [47, 48], but two additional reviews showed discrepancies using the Cochrane risk of bias criteria for some primary studies. Agreements and discrepancies among more than two SRs are shown in "Table 8".

## The generalisation of the evidence

"Table 9" summarises the generalisation of the evidence resulting from the discussion of the results from included reviews around a list of pre-specified relevant questions. We responded to these questions weighing up the overall bias from the included reviews (see "Table 4"), their overlapping (see "S4 Table") and the width of research which were synthetized ("Table 5").

**Is there evidence of a general autonomic effect of manual therapy interventions across reviews?.** Not all the included reviews assessed the same autonomic marker; therefore, evidence about a general effect of manual therapy interventions is not found among the reviews included. However, depending on the autonomic marker and system, an autonomic effect has been observed after manual therapy interventions.

**Has this effect a specific ANS direction (i.e. sympathetic, parasympathetic)?.** It seems consistent across reviews an SNS excitation when assessing skin autonomic activity. A PNS excitation seems to occur in cardiovascular autonomic activity, although it has not been mainly studied among included reviews.

**Is there evidence of the relation between the autonomic effect and the decrease of pain or any other clinical outcome?.** Five reviews included did not report the clinical relevance of the findings. The ones that reported it claimed caution when interpreting the clinical association between ANS effects and clinical outcomes improvement due to their inconclusive results.

**Is this ANS effect different when using an isolated MT technique or combining different MT techniques/approaches?.** Only one review [48] included studies with a combination of different MT and did not report the clinical relevance of the findings.

**Is the ANS effect robust across conditions amongst different types of interventions and age groups?.** A consistent effect across conditions and age groups has not been observed due to the reviews' heterogeneity. However, regarding the type of interventions, the SNS excitation found in skin autonomic activity can be elicited by mobilizations. The PNS activation of cardiovascular autonomic activity seems to be elicited when applying manipulations to the upper neck and lumbar spine and applying myofascial techniques.

**Is the body region where the technique is applied important to observe any specific effect?.** Some particularities were found among reviews regarding the body region where the manual technique was applied. A correlation between the upper neck and lumbar spine with a PNS activation and between the lower neck and upper thoracic with SNS activation has been

**Table 8. Agreements and discrepancies among reviews [more than 2] rating the same article.**

| Primary studies | Agreement among reviews rating the same article | Systematic reviews |
|---|---|---|
| Budgell and Hirano (2001) | NO | **Amoroso-Borges 2018, Araujo 2018, Picchiotino 2019, Rechberger 2019** |
| Budgell and Polus (2006) | NO | **Amoroso-Borges 2018, Araujo 2018, Picchiotino 2019, Wirth 2019** |
| Chiu and Wright (1996) | NO | **Schmid 2008, Hegedus 2011, Chu 2014, Kingston 2014, Araujo 2018, Navarro-Santana 2019, Rechberger 2019** |
| Chiu and Wright (1998) | NO | **Schmid 2008, Hegedus 2011, Navarro-Santana 2019** |
| Henderson et al. | NO | **Hegedus 2011, Picchiotino 2019, Rechberger 2019** |
| Jowsey and Perry (2010) | NO | **Hegedus 2011, Chu 2014, Kingston 2014, Araujo 2018, Navarro-Santana 2019, Picchiotino 2019, Rechberger 2019** |
| La Touche et al. (2013) | NO | **Chu 2014, Lascurain 2016, Araujo 2018, Navarro-Santana 2019, Picchiotino 2019, Rechberger 2019** |
| McGuiness et al. (1997) | NO | **Schmid 2008, Kingston 2014, Araujo 2018, Picchiotino 2019** |
| Moulson et al. (2006) | NO | **Hegedus 2011, Chu 2014, Araujo 2018, Navarro-Santana 2019, Picchiotino 2019** |
| Moutzouri et al. (2012) | NO | **Araujo 2018, Navarro-Santana 2019, Picchiotino 2019** |
| Paungmali et al. | YES | **Navarro-Santana 2019, Picchiotino 2019** |
| Perry and Green (2008) | NO | **Hegedus 2011, Kingston 2014, Picchiotino 2019** |
| Perry and Green (2011) | YES | **Araujo 2018, Navarro-Santana 2019** |
| Perry et al. (2011) | YES | **Amoroso-Borges 2018, Araujo 2018** |
| Petersen et al. (1993) | NO | **Schmid 2008, Hegedus 2011, Chu 2014, Kingston 2014, Navarro-Santana 2019, Picchiotino 2019, Rechberger 2019** |
| Piekarz and Perry | YES | **Navarro-Santana 2019, Picchiotino 2019** |
| Puhl et al. (2012) | YES | **Araujo 2018, Picchiotino 2019** |
| Roy et al. (2009) | YES | **Amoroso-Borges 2018, Picchiotino 2019, Rechberger 2019, Wirth 2019** |
| Sillevis et al. (2010) | YES | **Araujo 2018, Picchiotino 2019** |
| Simon | YES | **Navarro-Santana 2019, Picchiotino 2019** |
| Slater et al. (1994) | NO | **Hegedus 2011, Araujo 2018, Navarro-Santana 2019, Picchiotino 2019** |
| Sterling et al. (2001) | NO | **Schmid 2008, Kingston 2014, Lascurain 2016, Araujo 2018, Navarro-Santana 2019, Picchiotino 2019** |
| Tsirakis | NO | **Navarro-Santana 2019, Picchiotino 2019** |
| Vicenzino et al. (1994) | NO | **Schmid 2008, Chu 2014, Araujo 2018, Navarro-Santana 2019, Picchiotino 2019** |
| Vicenzino et al. (1998) a | NO | **Schmid 2008, Kingston 2014, Lascurain 2016, Araujo 2018, Navarro-Santana 2019, Picchiotino 2019** |
| Vicenzino et al. (1998) b | NO | **Schmid 2008, Chu 2014, Picchiotino 2019** |
| Ward et al. | NO | **Galíndez-Ibar 2017, Picchiotino 2019, Wirth 2019** |
| Welch, Boone (2008) | YES | **Amoroso-Borges 2018, Rechberger 2019** |
| Win et al. | YES | **Rechberger 2019, Wirth 2019** |
| Zegarra-Parody | YES | **Navarro-Santana 2019, Picchiotino 2019** |
| Zhang et al. (2006) | YES | **Amoroso-Borges 2018, Rechberger 2019** |

**Table 9. Outline of generalization of the evidence.**

| GENERALISATION | Wirth 2019* | Picchiottino 2019* | Araujo 2018* | Chu 2014* | Navarro-Santana 2019 | Hegedus 2011 | Rechberger 2019 | Galindez-Ibarbengoetxea 2017 | Kingston 2014 | Amoroso-Borges 2017 | Lascurain 2016 | Schmid 2008 |
|---|---|---|---|---|---|---|---|---|---|---|---|---|
| Is there evidence of a general autonomic effect of manual therapy interventions across reviews? | SNS/PNS | SNS | SNS | SNS | SNS | SNS | ANS | NO | SNS | PNS/SNS | SNS | SNS |
| Has this effect a specific ANS direction (i.e. sympathetic, parasympathetic) | SNS/PNS | SNS (skin) | SNS | SNS (SC/ST) | SNS (SC/ST) | SNS (SC) | ANS | NO | SNS (SC/ST) | PNS/SNS | SNS(SC) | SNS (SC) |
| Is there evidence of the relation between the autonomic effect and the decrease of pain or any other clinical outcome? | NO | NO | NR | YES | NO | YES | NR | NR | YES | NR | NO | NR |
| Is this effect different when using an isolated MT technique or a combination of different MT techniques/ approaches? | NR | NR | NR | NR | NR | NR | YES | NR | NR | NR | NR | NR |
| Is the effect robust across conditions, type of interventions and age groups? | NO | NO | NO | NO | NO | NO | NO | NO | NO | NO | NO | NO |
| Is the body region where the technique is applied important to observe any specific effect? | YES PNS (cervical and lumbar)/SNS (thoracic) | NO | NO | NO | NO | NR | NO | NO | NO | YES PNS (cervical and lumbar)/ SNS (thoracic) | NO | YES SNS (Cervical) |
| Is there evidence for the duration of the autonomic effect? (short, medium or long-term effect) | short | short | short | short | short | short | short | short | short | short | short | short |

H: High Risk of Bias in ROBIS assessment; U: Unclear Risk of Bias in ROBIS assessment; L: Low Risk of Bias in ROBIS assessment; SNS: sympathetic Nervous System; PNS: Parasympathetic Nervous System; ANS: Autonomic Nervous System; SC: Skin Conductance; ST: Skin Temperature; NR: Not reported.

*Low Risk of Bias (ROBIS Assessment).

consistently observed. However, most reviews with higher methodological quality [43, 45, 46, 49] could not differentiate a specific effect depending on the body region where the technique was applied.

**Is there evidence for the duration of the autonomic effect?.** *(short, medium or long-term effect)*. No evidence for a medium-term effect (from 1 to 24 weeks) or a long-term effect (more than 24 weeks) exists about the autonomic effect elicited by manual therapy interventions due to a lack of studies.

**Do the contextual factors and/or the non-specific effects of MT interventions influence the autonomic effects?.** None of the included reviews reported contextual factors and the non-specific effects of MT interventions.

**Can we infer that the effect might be observed across conditions that are not included in the current overview?.** Until more studies show the effect across different populations and conditions, no other inference can be made.

As planned, we attempted to perform a more thorough quantitative analysis of the included studies. However, the low number of meta-analyses included in the SRs and the high heterogeneity of the data prevented us from formally conducting further advanced analyses, including a meta-analysis of the data.

## Discussion

This overview included 12 systematic reviews on the autonomic effects of manual therapy interventions. Although there was a high degree of overlap between reviews, none were excluded to delve deeply into their high degree of heterogeneity.

Overall, the present overview of SRs suggests that MT techniques elicit changes in the ANS. Consistent results were found for a short-term SNS excitation of autonomic skin activity when assessed with SC after spinal mobilisations regardless of the body region treated. A PNS effect is seen when evaluating the ANS with HRV, despite some contradictory findings and that PNS autonomic markers have not been adequately studied in the reviews included in this overview.

Beyond the findings observed in the SRs, four primary areas of discussion emerged from a closer look at the results: 1) the specific effects following MT interventions (that includes the types of techniques and the body region of treatment) and the autonomic markers used to assess these effects; 2) the clinical relevance of the findings 3) the non-specific effects of MT interventions that might influence ANS, and 4) the methodological aspects related to MT research conducted in the field of ANS.

### Autonomic specific effects following MT interventions

The current overview shows that the effects elicited by MT on ANS rely mainly on the autonomic marker used. It is important to highlight that peripheral ANS has an extraordinary functional specificity defined according to the effector cells [74]. In this context, a general consistent bodily response is, indeed, improbable, and therefore, general activation or inhibition of the ANS is very unlikely [34, 51, 74, 75]. The following section discusses the different types of techniques, the related specific effects and the body region treated and categorised by the ANS marker used.

**Assessing ANS with skin conductance.** The reviews using SC reported a consistent acute sympatico-excitation effect independent from the technique used and the body region where the technique was applied.

Skin conductance can quantify the SNS state more reliably because of the sympathetic cholinergic system control on the sudomotor activity [76]. This fact might explain the specific activation of the SNS when assessing with SC. Interestingly, sudomotor activity is independent of

haemodynamic variability and respiratory rate and can be affected by skin quality, moisture levels and environmental temperature (see [77] for a comprehensive review). In addition, local physiological variations (e.g. skin temperature) due to touch or prolonged contact with the operator could influence the effects of manual interventions on the SC parameter rather than an increase in sympathetic activity "per se". For this reason, the SC assessment has to be carefully interpreted and especially when comparing between individuals focusing on each patient's variability.

**Assessing ANS with HRV.** Reviews using HRV concluded that spinal manipulations, mobilizations, myofascial and cranial techniques could elicit a short-term PNS activation. Body regions seem to influence when spinal manipulations are applied to lower cervical and upper thoracic areas, as one review found an SNS activation [47]. In addition, one SR did not find any effect on HRV after MT interventions [45].

HRV is a reliable marker to assess the autonomic nervous system [5, 6, 8, 78–80]. It consists of the change in the time interval between successive heartbeats [5]. It can be measured during 24 hours, short-term (5 minutes) or ultrashort-term (<5 minutes), using time-domain, frequency-domain or non-linear measures [8]. Most of the time-domain indices provide an assessment of vagal tone [81]. For the frequency-domain indices, HF is influenced by PNS activity, whereas the LF is influenced by both sympathetic and parasympathetic components making the method less specific for SNS activity; the ratio LF/HF reflects sympathovagal balance in terms of cardiac modulation [5–7, 81]. Non-linear measurements are considered sensitive parasympathetic indices [25]. Complex interactions between linear and non-linear measurements are necessary to interpret HRV properly. Indeed, an increased PNS activity may be associated with a decrease, increase, or no change in SNS activity [8]. As shown in the results, most of the articles using HRV measurements combined few HRV metrics, reducing the possibility of understanding the ANS reaction fully.

Two recent randomized clinical trials combined linear with non-linear metrics in the HRV measurement of different manual interventions confirming the probable PNS activation [25, 82, 83]. Furthermore, only one review [44] correlated the time with frequency domain indices and no one correlated linear with non-linear metrics. A recent SR aiming to investigate whether HRV parameters are altered in people with chronic low back pain compared to healthy controls suggested that it may be probable that chronic low back pain patients presented a lower vagal activity evidenced by HRV [53].

On the contrary, data obtained with parameters such as LF and LF / HF are not very specific and are not always an index of the autonomic balance [5, 8, 80]. When comparing frequency values as LF and HF among the studies, conclusions can not be accurate if values are not compared in the same units (absolute or normalized) because their interpretation might differ [81].

Besides, any review included long-term HRV measurement, representing the gold standard for clinical HRV assessment [8]. Moreover, Laborde et al. proposed the 3R of HRV (resting, reactivity and recovery), measuring HRV at three points: baseline, event and post-event. Therefore, accurate analysis of HRV is necessary to estimate more precise and reliable results [4, 5, 8, 74, 76].

**Assessing ANS with other markers.** Other markers used to measure the effects on ANS were ST, SBF, RR, HR, BP, pupillometry and plasma catecholamines. When assessing SBF and ST, contradictory results were found. A decrease of ST (without specifying where the ST was measured) was found after mobilisations of different body regions in most of the SRs unless for Picchiottino et al. reporting no effect [45]. Indeed, both an SNS activation and a PNS activation has been shown.

Interestingly, Zegarra-Parodi et al. pointed out that non-sympathetic factors are also involved in SBF and ST regulation (e.g. local response of vascular endothelial cells to

metabolites and environmental and humoral stimuli), questioning, therefore, the appropriateness of such markers assessing the sympathetic cholinergic system [42, 84]. Indeed, its interpretation could be more accurate when correlating the results with other autonomic markers. The gold-standard method for measuring sympathetic activity is through direct intraneural measures using microneurography, although other rigorous methods, including norepinephrine spillover techniques, can be successfully employed [85].

In our overview, several articles [41, 45, 47–49, 70–72] assessed ANS activity with BP and HR, and contradictory results were also found among the studies. A decrease in BP was found after cranial techniques and an increase after spinal manipulations to the upper neck. In addition, no effect was found after spinal mobilizations. Many mechanisms can influence BP, depending on which factor is driving the blood pressure response. An increase in blood pressure can affect the modulation of autonomic outflow via the baroreflex [80, 86]. Thus, only cautious interpretations can be made. It is worth noting that blood pressure, heart rate, and respiration are the primary physiological parameters that could be recorded during autonomic tests, from which heart rate variability and blood pressure variability, among other autonomic parameters, are calculated. Therefore, they must be considered to contribute equally and to have the same importance when measuring, assessing and interpreting the overall autonomic response [6, 78, 79, 87].

Surprisingly, only a few reviews reported the RR assessment, concluding an increase after mobilisations. The assessment of RR is fundamental when exploring ANS on the cardiovascular system because it influences on HR [79, 87]. However, RR can be affected by age, resting heart rate, body mass index and pharmacological assumptions [6].

Three SRs [45, 47, 49] included primary articles assessing ANS through pupil diameter but did not find any effect after MT interventions. Interestingly, Araujo et al. noted a different response of the pupil diameter when comparing the intervention with a placebo instead of no intervention, suggesting a potential non-specific effect of placebo on ANS [49]. The use of pupillometry to assess MT interventions effects is rare compared to other ANS markers. As pointed out by some authors, it is not easy to interpret that it requires specialised expertise and skilled specialists are needed to perform it correctly [79].

The SRs using plasma catecholamines to assess ANS activity found no effect when applying cranial techniques and spinal manipulations [45, 48]. Plasma catecholamines is a biochemical indicator of the sympathetic adrenergic system (dopamine, adrenaline, noradrenaline) but have some limitations: noradrenaline is subject to reuptake, its proportion circulating is very small related to all the amount of neurotransmitters secreted from nerve terminals, and it does not allow an assessment of PNS or sympathetic cholinergic system because plasma acetylcholine is highly labile and quickly decomposed preventing quantification [6, 7, 76]. Therefore, it should be used in combination with other autonomic markers to strengthen the interpretation of the results [7].

**Autonomic markers: the need for simultaneous multiple autonomic measurements.** The methods used to assess the ANS differ among the reviews included. The most common markers were HRV and SC, as confirmed by the ANS literature [79]. Although several indices are recommended to study the ANS response more in-depth [45, 80, 87], most articles assessed the ANS partially. Because of the complexity of the autonomic system, it has been argued that no single test can show a high level of accuracy for specific elements and functions of the ANS [6, 7, 76, 80]. Indeed, the combination of complementary measures, such as HRV and SC as well as variations on BP [87] and HR during head-up tilt test, Valsalva manoeuvre and deep breathing [76], is essential to cover the different ways in which the ANS responds to external stimuli and to obtain a more precise and realistic picture of the ANS function [25, 76]. Also, the Autonomic Reflex Screen is recommended to test cardiovagal, sudomotor and adrenergic

functions of the ANS in a standardized fashion way [80]. Therefore, the reduced use of simultaneous multiple autonomic measurements might limit the quality of the results.

In addition to taking more than one measure, it is necessary to consider that the information given by those markers can be influenced by several physiological and psychological conditions [5, 30, 75, 78]. Stable variables (i.e. age, sex, toxic habits, medication) and transient variables (routines as sleeping, physical activity and meals) should be routinely collected while conducting ANS research to guarantee validity and reproducibility [6, 7, 79]. None of our included reviews reported baseline variables of the participants they included.

## Clinical relevance of the autonomic effects elicited by MT interventions

Even if most of the reviews included reported data about symptomatic and asymptomatic populations, many of the studies included in the reviews were conducted on healthy patients. So, the translation into a real clinical-based practice setting should be done with caution. Moreover, most of the articles reviewed assessed only short-term effects, and very few used additional PROMs to correlate with ANS changes. These elements raise doubts about the clinical relevance of these results: an argument also initially discussed by Picchiottino et al. and Wirth et al. [45, 47].

In clinical practice, most patients seeking manual interventions have chronic diseases [16–18, 88]. Up to now, scientific literature has described only the short-term effects of MT [63], so exploring the sustained effects through sessions is crucial. Nevertheless, it has been argued that short-term effects might not have a predictive value over time [25], implying the need for assessing both short- and long-term effects [50].

Initial evidence showed contradictory results for SC and pupillometry studies correlating ANS indices scores with pain values [78, 89]. Indeed, Lascurain-Aguirrebeña et al. did not find any correlation between sympathetic-excitatory changes and neck pain relief [90]. Notwithstanding this, Carnevali and colleagues reported preliminary evidence of the beneficial effects of MT in three pilot studies conducted on pathological (e.g. hypertension) and physiological conditions (e.g. recovering from sports competition) [50]. Therefore investigating the impact of MT on the autonomic indices enrolling patients with pain or other clinical conditions, including dysautonomic syndromes, is paramount.

## Non-specific ANS effects after manual therapy interventions

A relevant point to consider when assessing ANS after MT interventions are the number and type of control groups, namely the effect of placebo response in ANS studies. The placebo/sham effect (caused by expectation and learning) affect on the central nervous system eliciting a hypoalgesic reaction by modulating the autonomic response [91–93]. Interestingly, Navarro-Santana and colleagues argued that the effect size of real treatment, e.g. SC and ST, is reduced from large to moderate when joint mobilisations were compared to a placebo group rather than a hands-off control group [46]. Furthermore, Araujo et al. found that the pupil diameter did not change after manipulations compared to placebo but decreased compared to no-treatment [49]. These results endorse the fact that placebo/sham can elicit neuropsychological effects (i.e. conditioning or expectation), modifying the body-brain interactions through ANS, among other systems. For a comprehensive review, see [91–97]. Three-quarters of the articles included in this review were sham-controlled clinical trials; however, different placebo/shams were used as a control group. This fact implies high methodological heterogeneity among studies in terms of the type of placebo used that might, in part, explain the inconclusive and contradictory results found in our review.

Moreover, contextual factors (during therapeutic encounters between patients and health providers) in manual therapy interventions are crucial to consider because of the influence on

ANS [98]. These factors can modify the autonomic response and might modulate pain experience [98]. Also, the emotional/cognitive state of the subject is important to take into account as it can affect autonomic responses [99].

The use of appropriate, adequate methodological protocols is even more relevant if the interpretability, robustness and validity of the results are taken into account: indeed, this is linked to the type of control groups used in the studies, the associated clinical measures utilised, and the type of population enrolled. Some authors have already shed light on the inconsistency of results when clinical conditions (i.e. pain) are considered. [30, 77, 78].

## Quality of the evidence and methodological considerations of ANS research on MT interventions

The quality of the included review was assessed by the ROBIS tool. This tool is highly recommended when performing overviews or clinical guidelines [65].

We have provided a detailed, transparent assessment of the quality of included reviews describing the rationale related to the ROBIS questions ("S4 Table"). The difference between those with low risk of bias and those with unclear risk of bias was due to the poor reporting of information, lack of protocol registration and methodological deficiencies (poor data analysis and reporting of the data, limited answer to the research question, absence of reporting search strategy).

Different methods to assess the quality or the risk of bias have been used within the included SRs. We observed discrepancies in the judgement to some studies included in several reviews; in part for the different approaches to this construct by the different scales and domain-based approaches, but also related to other factors such as a variance due to learning curve or the knowledge of these tools. The quality of RCTs is a persistent issue in the field of MT. Several studies have shown a low quality of RCTs performed in MT [11, 100, 101]. Our overview identified a high heterogeneity in regards to the quality of the articles included. Moreover, recent studies investigating disagreements on rating the quality of RCTs included in more than one SR showed that the scores differed substantially where different reviews rate the same article differently [102, 103]. Another reason that leads to heterogeneity relates to the incomplete or unclear reporting, as also noted by Alvarez et al. [11]. Consistent with the literature, the studies included in the present overview revealed the same issues, such as high heterogeneity, reporting deficiencies and the same articles rated differently.

## Results from recent RCT's

Few RCTs have been recently published that were not included in the reviews included in this overview.

Cerritelli et al. aimed to study the ANS effects after osteopathic manipulative treatment. They combined different autonomic markers to assess the autonomic response of MT interventions concluding a PNS activation in the cardiovascular assessment and the thermal skin response and an SNS activation in skin conductance [25]. In addition, these effects were maintained at medium-term. This PNS activation on cardiovascular assessment was confirmed by four other studies [82, 83, 104, 105]. All of them include different types of techniques applied in different body regions.

## Suggestions and recommendations for further research

Basic research has evolved our knowledge concerning ANS function. However, daily care is a complex environment, where the complexity of the patients requires complex multidimensional interventions. This complex scenario pushes science to extend the research from bench

to bedside to enactivism [106–108]. In that sense, designing RCTs comparing an intervention with placebo and no treatment, or assessing ANS markers in patients and correlating the results with patients benefits, in a more ecological environment, could fill the gap of clinical transferability [109]. In terms of design, different methodological solutions have been proposed recently to adapt RCTs for complex interventions like MT. For example, realistic RCTs aim to assess the effects and the mechanisms behind this effect and even how these mechanisms interact with the context [110]. Pragmatic RCTs and comparative effectiveness of N-of-1 trials could also be suitable alternatives [111, 112], especially when the high variability among individuals is an issue, and thus the solution is to focus on individual variability [77].

According to our results, we argue that several points should be taken into account when performing research exploring the autonomic effects of MT interventions ("Table 10").

First, to reach a robust conclusion on the effects of a given MT intervention, several autonomic markers (assessing different body domains) should be considered in the study design. Combining various measures is recommended to obtain a clear picture of ANS activity [7, 25, 45, 87, 113]. For example, a combination of HRV, SC, thermography, BP and RR could improve the assessment of ANS. Moreover, complementary assessments using PROMs about autonomic symptoms could help evaluate the relevance of the results offered by the autonomic markers [77]. Outcomes should be able to relate ANS changes with therapeutic benefits.

Although the research on healthy subjects could bring preliminary results, studies including specific populations (e.g. chronic pain patients) would increase the clinical relevance. Additionally, correlating the results with the anamnesis and the physical examination might increase the pertinence of the physiological outcomes. In addition, controlling for confounding factors is essential: for example, caffeine and nicotine should be abstained from at least 3

**Table 10. Suggestions and recommendations for further research in the field of autonomic nervous system.**

| Topic | Current issue | Proposed solution | Expected impact | Example in manual research |
|---|---|---|---|---|
| **Autonomic markers** | Use of a single marker | Contextual use of multiple autonomic methods | Broaden the study of the ANS function | Use of the following methods: HRV, SC, thermography, BP and RR |
| **Autonomic assessment** | Uncertain relevance of the results | Broaden the assessment to complement the ANS markers | Correlate physiological outcomes with clinical relevance | Use of PROMs, anamnesis and physical examination |
| **Confounding factors** | Heterogeneity of the results and lack of replicability | Standardise the variables that can influence the ANS assessment | Replicability of the studies and methodological homogeneity | See example guidelines Laborde et al. and Zygmunt et al. |
| **Population** | Research mostly in healthy subjects | Include specific population and comorbidities | Increase clinical relevance | Include patients with chronic pain |
| **Follow-up** | Short-term effects of MT | Include longer time points | Long-term effects of MT | measures at 5, 15, 30, 60, 120 min after the MT |
| **Quality of the studies** | Poor reporting | Use of design and reporting tools | Increase of the quality | Use of Tidier, Consort, Precis |
| **Study design** | Doubtful applicability of the results | New approaches and study designs | Clinical relevance and applicability of results | Realistic RCT |

ANS: Autonomic Nervous system; HRV: heart rate variability, SC: skin conductance; BP: blood pressure; RR: respiratory rate; PROMs: patient reported outcomes; MT: manual therapy; RCT: randomized clinical trials.

hours before testing, no physical training the day before, no meal 2 hours before the intervention, no alcohol for 24 hours before the measurement and where possible drugs affecting ANS should be avoided at least 48 hours before the study begins. Moreover, the patient should use the bathroom before the intervention and be laid down or seated for about 30 minutes in a quiet room with a neutral temperature and humidity [5, 6].

Regarding the research design and reporting, MT studies on the ANS field should consider several points: long-term autonomic effects of MT interventions have to be explored. The use of standardised intervention reporting guidelines should be thoroughly recommended to improve study quality [102, 103, 114]. Also, articles should describe interventions more appropriately to adequately understand the techniques used in the studies, even if a patient-based approach is applied. Finally, new designs and experimental approaches can be considered in the complex scenario of MT.

## Key messages

- *Research suggests that MT can produce both a PNS and an SNS short-term effect depending on the ANS measurement method.*

- *Skin conductance and heart rate variability are the most frequent autonomic measures to assess ANS. However, there is the need for a combination of measures to have a clearer and a more robust conclusion about the effect of MT interventions on ANS*

- *There is no evidence of the association of specific manual techniques with precise autonomic responses*

- *There is still lack of robust evidence in regards to the body regions where the technique is applied and specificity of the autonomic effects*

- *Based on all current evidence, it is difficult to conclude whether the findings on the autonomic effects produce any clinically relevant results in these studies*

### Potential biases in the overview process, strengths and limitations

None of the authors of this overview was involved in any of the reviews included in the primary studies. We used an unambiguous definition of systematic reviews [57] to establish eligibility criteria and ensure the inclusion of studies conducted rigorously and that could identify the breadth of evidence informing the clinical question. To identify the completeness of existing reviews we planned a search strategy adhering current guidance [115]. We aimed to obtain a scoping picture from the reviews addressing the potential autonomic effects of manual therapies resulting in broad eligibility criteria to include in a first step review irrespective of their publication date or quality. Current guidance to conduct overviews does not reach a consensus to recommend an approach to assess the risk of bias from systematic reviews when conducting an overview [54, 55]. Although AMSTAR and ROBIS have shown similar reliability and performance [116, 117] we made the decision to use ROBIS as it allows a domain-focused appraisal. The decision to include any identified reviews, data extraction and methodological quality assessment of the included reviews was based on an independent assessment by two overview authors (SR, GA) with discussion involving a third overview author (FC) when a disagreement arose.

Although we recognise that potential biases exist at all stages of the overview process, efforts have been made to reduce them throughout the process, especially in the generalisation of the evidence, where no validated tool has been used. Some of these biases were inherent to the

characteristics of included reviews but we established mechanisms to address them analysing the overlap between reviews and with a formal assessment of their bias, allowing us to interpret their findings accordingly. Also, the recency of some reviews [41, 43, 72, 73] could be a source of concern, but we formally assessed their up-to-dateness [118] and interpreted their results in consequence. It would be informative to have included judgements on the certainty of our findings, but at present formal guidance on how to use GRADE in conducting overviews is still pending [54].

### Implications for research

According to all the issues discussed above, our findings have some implications for research and allow suggesting specific recommendations in order to improve both the quality and clinical relevance of future studies in this field. Future studies should be designed with pragmatic approach and promote the inclusion of individuals with specific conditions. The impact of manual therapies should be assessed using a variety of standardised autonomic markers in order to cover the entire range of potential ANS effects. Also, the studies should improve their design and execution considering patient-centred approaches, selecting control conditions carefully, anticipating the effect of confounding and planning a follow up at long term. Finally, a complete, accurate and transparent report of all the studies conducted should be ensured.

### Conclusions

The findings of this overview of systematic reviews showed that manual therapies may have an effect on both sympathetic and parasympathetic systems. However, the systematic reviews included showed inconsistent results, largely explained by differences in their methodological rigour and how the effects were measured. The majority of reviews with a lower risk of bias could not discriminate the effects depending on the body region to which the technique was applied. In consequence, the magnitude of the specific autonomic effect elicited by manual therapies and its clinical relevance is uncertain.

### Supporting information

**S1 Checklist.**
(PDF)

**S1 Table. Search strategy.**
(DOCX)

**S2 Table. Data extraction form.**
(XLSX)

**S3 Table. Risk of bias of included reviews (complete ROBIS assessments).**
(DOCX)

**S4 Table. Concordance overlapping and conclusions.**
(DOCX)

**S5 Table. Agreements and discrepancies.**
(DOCX)

### Acknowledgments

We would like to thank Andy Otaqui, Luca Carnevali and Prof. Andrea Sgoifo for their help in reviewing the paper.

## Author Contributions

**Conceptualization:** Sonia Roura, Gerard Álvarez.

**Data curation:** Sonia Roura.

**Formal analysis:** Sonia Roura.

**Funding acquisition:** Sonia Roura.

**Investigation:** Sonia Roura, Gerard Álvarez.

**Methodology:** Sonia Roura, Gerard Álvarez, Ivan Solà, Francesco Cerritelli.

**Supervision:** Gerard Álvarez.

**Validation:** Sonia Roura, Francesco Cerritelli.

**Writing – original draft:** Sonia Roura.

**Writing – review & editing:** Gerard Álvarez, Ivan Solà, Francesco Cerritelli.

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
