## [Decision Letter · Decision Letter 0]

6 Sep 2021

PONE-D-21-24377Do manual therapies have a specific autonomic effect? An overview of systematic reviews.PLOS ONE

Dear Dr. Roura,

Thank you for submitting your manuscript to PLOS ONE. After careful consideration, we feel that it has merit but does not fully meet PLOS ONE’s publication criteria as it currently stands. Therefore, we invite you to submit a revised version of the manuscript that addresses the points raised during the review process. Having intensively reviewed your revised draft, our external reviewers differed with their final recommendations, at least to some extent. Thus, I have double checked your revised version, to come to a more balanced decision (see R #1). All in all, our  identified shortcomings are considered reasonable with regard to both PLOS ONE’s quality standards and our readership's expectations. Therefore, we invite you to submit a carefully revised version of the manuscript that addresses EACH AND EVERY point raised during the current review process. Please note that a non-convincing revision (not considered acceptable with regard to language, content, reviewers' constructive criticisms, generalizable conclusions, and/or Authors' Guidelines) must lead to outright reject. 

We look forward to receiving your revised manuscript.

Kind regards,

Andrej M Kielbassa

Academic Editor

PLOS ONE

Journal Requirements:

 [Funding for the publication of this article was provided by Registro de Osteopatas de España (ROE) www.osteopatas.org].  

Reviewers' comments:

Reviewer's Responses to Questions

**Comments to the Author**

1. Is the manuscript technically sound, and do the data support the conclusions?

Reviewer #1: Yes

Reviewer #2: No

Reviewer #3: Yes

2. Has the statistical analysis been performed appropriately and rigorously? 

Reviewer #1: Yes

Reviewer #2: N/A

Reviewer #3: Yes

3. Have the authors made all data underlying the findings in their manuscript fully available?

Reviewer #1: Yes

Reviewer #2: Yes

Reviewer #3: Yes

4. Is the manuscript presented in an intelligible fashion and written in standard English?

Reviewer #1: Yes

Reviewer #2: Yes

Reviewer #3: Yes

5. Review Comments to the Author

Reviewer #1: This is a review of systematic reviews. I have no comments to the authors. The only thing is checking the file uploading since tables are mixed with text and it has been difficult to review the paper. Authors could explain why they have not conducted an umbrella meta-analysis.

Reviewer #2: General remarks

- In formulating the scope for a review of reviews, the PICOS (participants, interventions, comparators, outcomes, and study design) structure is considered helpful. Please revise accordingly.

- Please note that there are various evidence levels with recent research papers. Consequently, a Systematic Review of Level I would include only Level-I Randomised Controlled Studies. No doubt, a Level-I Overview of Systematic Reviews must include Level-I Systematic Reviews only. It would not make sense to re-repeat poor RCTs, and to re-repeat poor Systematic Reviews based on poor RCTs.

- Please remember that Plos One's mission is "to publish methodologically and ethically rigorous research". Would it be sound from a methodological point of view to include papers based on poor quality? No, it wouldn't. Would it be sound from an ethical point of view to draw any conclusions based on such papers? Again, no, it wouldn't, too. But this clearly would render your "rigorous" research doubtful.

- What about using the AMSTAR tool as a means to assess the methodological quality of systematic reviews?

- What about using the GRADE approach with your ‘Summary of Findings’?

Abstract

- Please note that the allowed maximum word count with this section is 300. Please shorten considerably.

- Note all repositories have been used. Please provide reasons.

- "Two overview authors independently applied the selection criteria (...)." Please define the "selection criteria".

- "Our search identified 557 records, from which we included 12 reviews." No word about the quality of those 12 papers? Remember that just including 12 retrievable papers would not seem satisfying.

- "Moreover, the clinical relevance of those results are still unclear." Reasons remain unclear.

- "Future research should consider some key elements proposed to overcome common shortcomings and include ways to improve the quality and applicability of the results." This is not considered a conclusion referring to your aims; instead (as with most other topics), this would have been clear prior to the start of your project.

Intro

- Again, please stick to the Authors' Guidelines. "(...) body’s internal environment (homeostasis) (1,2)." must read "(...) body’s internal environment (homeostasis) [1,2]." Revise thoroughly.

- "Recent systematic reviews have shown that the MT approach is clinically effective (...)." and "iMoreover, several clinical studies have shown the effectiveness of MT (...)." Please see comments given above, and add information on the quality level of the referenced papers. Just "being published" is not considered a quality aspect.

- Same with "Evidence about autonomic effects of manual therapy interventions is synthesized within many reviews (41–46,48,49)." Again, provide detailed information, and revise carefully.

Meths

- Did you note that your "Objectives" (aims) do not fully correspond to your Abstract section?

- regarding your inclusion criteria, please see comments given above. Revise carefully, and include aspects of utmost quality.

- "Cochrane Library, PubMed, EPISTEMONIKOS, and SCOPUS" would not seem exhaustive. Why didn't you include other databases? What about foreign languages? What about the grey literature?

- "Two authors (SR and GA) independently assessed titles and abstracts of records identified by the electronic searches according to the inclusion criteria and decided on eligibility obtaining a full text copy from relevant references." Indeed, the two authors obviously simply searched. Again, what about the quality level of the included papers?

- "We solved disagreements involving a third author (FC) to reach a consensus through discussion." What kind of "disagreements" are you talking about? How often have there been "disagreements"? Was a consensus possible in all cases? And, again, what about the agreements regrading the quality levels?

Results

- "All the reviews included randomized clinical trials (RCTs), one also included quasi-RCTs, four included non-randomized trials and two other designs." Again, see comments given above. "Quasi-RCTs, non-randomized trials and other designs" would not confirm Level-I research. Why did you include such papers?

- With your Table 2 ("Characteristics of the included reviews"), please add your quality assessment. Again, please do not simply add the published papers; instead provide a sound, a valid, and a reasonable quality assessment, to allow to distinguish high- from poor-level papers.

- Do not use Authors' names with your text, see "Schmid et al. found conflicting evidence (69).".

- There would seem some errors to be evident. See "and after mobilisations (41,45) and in the diastolic blood pressure after neck manipulations (Schmid et al. 2008)."

- Same with "Galindez et al. found a decrease in systolic BP in hypertensive type I subjects but not in healthy participants (Galindez et al 2016). Two studies found an increase in HR after mobilisations (49,67), and one found no effect after cervical manipulation (Galindez et al 2017)." revise carefully.

- "Different quality assessment tools were used among the SRs (Appendix 4), showing a high heterogeneity of the studies' quality." See comments given above. First, the quality of the reviews would seem more important than the contents, to decide whether the paper will be included. Second, the results of your quality assessment must be provided here, and NOT somewhere in an Appendix.

- From your Table 9 (!) it finally becomes clear that you ONLY found 1 (!) paper of "high quality". This would not seem convincing, not at all.

- Revise thoroughly for minor typos, see ""(...) are considered. (30,76,112)."

- Again, do not use Authors' names, see "This tool is highly recommended when performing overviews or clinical guidelines (Whiting et al.)."

- "The difference between those with low risk of bias and those with unclear risk of bias was due to the poor reporting of information, lack of protocol registration and methodological deficiencies." OK, but this would not seem an acceptable excuse.

- "Our overview identified a high heterogeneity in regards to the quality of the articles included." Again, this would mean that papers with poor quality have been included. This must be elaborated more clearly. Moreover, it does not make sense to re-re-repeat those poor quality papers. With your revision, please stick exclusively to the Level-I papers. This refers to your conclusions. Papers with poor quality might be mentioned, but the drawbacks must be elucidated clearly.

- "Consistent with the literature, the studies included in the present overview revealed the same issues, such as high heterogeneity, reporting deficiencies and the same articles rated differently." So please explain why you have included those papers? What about excluding such RCTs/systematic reviews? Remember that your write for the readers. What should a reader think after having swallowed your overview?

- Please separate your "Suggestions" from the "Conclusions".

- Do not mix "Conclusions" and "Recommendations". All these aspects might be right, but do not mix it up, please.

Refs

- Stick exclusively to the Journal style, and revise for uniform formatting. Again and again, you state "[Internet]" and "Available from"; please delete. Consult some recently published Plos One papers. Provide doi and PMID numbers.

Concl

- With your Conclusions, please stick exclusively to your revised aims. Do not simply repeat your results here. Do not give a further literature review here. Instead, provide a reasonable and generalizable extension of your outcome.

All in all, this submitted draft would seem interesting, is considered easily intelligible and would seem worth following after a thorough revision, considering all the aspects indicated above.

Reviewer #3: I congratulate the authors for their work. The review addresses a relevant topic with important gaps in the literature. The compression of the mechanisms of effect of manual techniques is fundamental for the most adequate clinical reasoning and for the scientific advance in this subject. In fact, the relationship between autonomic effects and clinical improvement is an important gap in the literature. The large number of reviews on the subject can confuse readers and the authors knew how to unify the main findings of the studies in a very adequate way. However, I suggest better synthesizing the ideas in the introduction and discussion to reduce the scope of the work. Below are other small suggestions

Intro:

1. Overall the introduction is very complete, but excessively detailed and long. Theoretically, readers interested in the topic should already have a basic knowledge of the topic, or they can consult the bibliographic references for further details. I suggest reducing paragraphs, or even unifying some (the second and third could unify action mechanisms - and gaps in these mechanisms - of manual therapy, without the need for so many explanations about the definition and clinical conditions that this area usually deals with - theme of the second paragraph )

Methods:

2. The authors describe that they considered the level of evidence described in the reviews included, as well as the methodological quality of the studies included in the reviews, for the generalization of evidence. However, the criteria used to summarize the level of evidence of the findings were not clear. I suggest defining a better instrument for this, something like the GRADE approach or similar.

Results:

3. Due to the amount of information (due to the large number of studies and variables), I suggest deleting the "newspaper" column from Table 2.

4. Table 5. Review by Araujo et al, it seems that they would be "Intervertebral mobilisations" or "mobilisations" only, and not both. In this same table, some words in the techniques column are capitalized and others are not.

Discussion:

5. As the introduction, the discussion is too long. Due to the number of studies and research questions, the results are already extensive. I suggest summarizing the main topics both in the introduction and in the discussion. Many discussion topics have already been covered in the results. I suggest organizing the discussion more broadly (without so many sections). I suggest organizing the discussion according to the following topics: main findings; strenghts and limitations; comparison with previous studies; meaning of the study (clinical message and future directions).

6. I see no reason for the authors to report results from more recently published RCTs….out of the scope of the study.

6. PLOS authors have the option to publish the peer review history of their article (what does this mean?). If published, this will include your full peer review and any attached files.

Reviewer #1: No

Reviewer #2: No

Reviewer #3: **Yes: **Francisco Xavier de Araujo

---

## [Author Response · Author response to Decision Letter 0]

14 Oct 2021

Comments Responses

Rev#1

This is a review of systematic reviews. I have no comments to the authors. The only thing is checking the file uploading since tables are mixed with text and it has been difficult to review the paper. Authors could explain why they have not conducted an umbrella meta-analysis

 Thank you to Rev#1, the tables were included directly after the paragraph in which they were first cited, as suggested by the PLOS ONE author guidelines. However, we agree that it is difficult to review the paper with the tables in the middle of the text due to their size and quantity. We updated the manuscript and included all tables at the end. 

The choice to perform an overview instead of an umbrella meta-analysis was because:

1) The goal was to map, synthesise and explore discrepancies in the available systematic reviews

2) The scope of the research question is broad, with high heterogeneity among participants, outcomes and interventions

3) Because our objective is consistent with the one stated in Cochrane Handbook chapter V for Overview:: “The primary reason for conducting Cochrane Overviews is that using systematic reviews as the unit of searching, inclusion, and data analysis allows authors to address research questions that are broader in scope than those examined in individual systematic reviews and in cases where it is important to understand the diversity present in the extant systematic review literature”

Rev#2

General comment to “General remarks” from Rev#2

We are grateful for the effort made to review our manuscript and for highlighting some points that have contributed to improving the reporting of our findings.

We also would like to open a discussion about the approach of the reviewer to the overview and the scientific literature summarized in the study. The reviewer focuses many of his / her comments on levels of evidence and the appropriateness to consider ‘poor quality’ (sic) studies when conducting an evidence synthesis. The former argument led the reviewer to question if our overview has been conducted according to a sound methodology (and fit, in the end, with the PLOS ONE mission).

In that sense, we submitted our manuscript to PLOS ONE being aware of its mission and instructions for (candidate) authors. Moreover, we submitted the manuscript convinced that we accomplished with the most recent standards for conducting overviews (summarised in Chapter V from the Cochrane Handbook for Systematic Reviews of Interventions or in Gates M doi: 10.1186/s13643-020-01509-0). We conducted the overview according to these methodological standards and will refer to them in responding to most of the reviewer’s comments.

It is necessary to disclose that we consider the discourse around ‘levels of evidence’ outdated. Classifications and discussions based on levels of evidence are restrictive and, due to its focus on study designs and hierarchies, does not allow to analyse additional issues such as the consistency of a body of evidence, its breadth or the overlap of its scope (some has provided more discussion on this issue (doi: 10.1136/bmj.c4875)). At the very end the focus on levels of evidence can result in an epistemological bias that confers confidence in the goodness of a determinate study design (or, more worrying, to its position in a hierarchy).

Recently the science of evidence synthesis has evolved to a more comprehensive approach to methodological challenges related to the necessity to deal with the limitations from the existing literature (and their implications to future research and practice) or allowing researchers to analyse the overlap within existing reviews.

Specifically, overviews are an excellent tool to summarise the breadth of research to obtain a picture of the overall completeness and applicability of the body of evidence included in the original reviews. Researchers have the commitment to accurately assess, analyse and interpret the threats for the validity of the primary studies included in eligible reviews and are responsible to present the findings of their overviews warning readers, when necessary, about the limitations of the existing evidence.

The reviewer expressed his / her concern on “what should a reader think after having swallowed [our] overview”. We sincerely hope that the arguments above could contribute to clarify the methodological framework of our overview. 

 In formulating the scope for a review of reviews, the PICOS (participants, interventions, comparators, outcomes, and study design) structure is considered helpful. Please revise accordingly.

 Thank you for this comment; A better structure for the scope of the overview has been added in the Inclusion Criteria section page 6 lines 159-209

Please note that there are various evidence levels with recent research papers. Consequently, a Systematic Review of Level I would include only Level-I Randomised Controlled Studies. No doubt, a Level-I Overview of Systematic Reviews must include Level-I Systematic Reviews only. It would not make sense to re-repeat poor RCTs, and to re-repeat poor Systematic Reviews based on poor RCTs.

 We provide a detailed rationale for this issue at the “General comment to “General remarks” from Rev#2”.

However, we have modified table 2 (characteristics of the reviews included), table 4 (Risk of bias of included SRs), table 5 (Summary of findings) and table 9 (Generalization of the evidence) and the main text to make more clear rationale:

Main text page 11 line 375-382, 477-497

Table 2 has been modified in order to clarify the methodological characteristics of the included SRs. Page 43

Table 5 “summary of findings”: we have merged tables 5 and 6 adding a column with the GRADE assessment made by the SRs. Page 50 

Table 9. Outline of generalization of the evidence, summarizes the questions that this overview wants to answer within each SR taking into account the methodological quality assessed by ROBIs, a coloured column in green has been added to improve the visibility of the importance given to the higher quality SRs included for answering the questions. Page 61 

Please remember that Plos One's mission is "to publish methodologically and ethically rigorous research". Would it be sound from a methodological point of view to include papers based on poor quality? No, it wouldn't. Would it be sound from an ethical point of view to draw any conclusions based on such papers? Again, no, it wouldn't, too. But this clearly would render your "rigorous" research doubtful.

What about using the AMSTAR tool as a means to assess the methodological quality of systematic reviews?

 Current guidance to conduct overviews does not reach a consensus on the ideal tool to appraise the validity of included reviews (doi: 10.1186/s13643-020-01509-0). Many published methodological frameworks to conduct reviews mention AMSTAR, but most recent guidance emphasises on ROBIS.

Chapter V from the Cochrane Handbook for Systematic Reviews does not recommend one tool over another due to the lack of empirical evidence about their performance (https://training.cochrane.org/handbook/current/chapter-v#section--4). Besides, Cochrane provides specific guidance to use domain-based tools when assessing the risk of bias of included studies in a review. As ROBIS is conceived from this approach (domain-based tools) we made the decision to use this tool.

A sentence has been added to clarify this point. Page 24 lines 860-865

What about using the GRADE approach with your ‘Summary of Findings’?

 We agree that including judgements on certainty of evidence would be informative, but there is no formal guidance to use GRADE for overviews. Besides, GRADE encourages the assessment of outcomes that have a direct impact on decision making and our clinical question could be, in part, out of its scope.

We have included a comment in the discussion related to this issue .Page 24 lines 877-879

Abstract

Please note that the allowed maximum word count with this section is 300. Please shorten considerably. Thank you for this comment, Abstract has been shortened considerably

Note all repositories have been used. Please provide reasons. Recent data show that the best combination to identify SR in the context of overviews is to search MEDLINE and Epistmonikos complemented with reference checking (DOI: 10.1186/s12874-020-00983-3). 

We complemented this guidance with searches in scopus and WoS (the former, to track citations to relevant reviews).

We added a sentence at the ‘Potential biases …’ section (page 24 lines 854-858) to remark the exhaustiveness of the search strategy.

"Two overview authors independently applied the selection criteria (...)." Please define the "selection criteria" Selection criteria are already defined in the abstract, page 2 lines 27-29

"Our search identified 557 records, from which we included 12 reviews." No word about the quality of those 12 papers? Remember that just including 12 retrievable papers would not seem satisfying. This is an important point. Amended: The sentence “Five out of 12 SRs were rated as low risk of bias when assessed with ROBIS tool.” has been added to the Abstract Results section page 2 line 33

"Moreover, the clinical relevance of those results are still unclear." Reasons remain unclear. Amended, page 2 lines 38-41

"Future research should consider some key elements proposed to overcome common shortcomings and include ways to improve the quality and applicability of the results." This is not considered a conclusion referring to your aims; instead (as with most other topics), this would have been clear prior to the start of your project. We agree with the reviewer that suggestions should go separately from conclusions. This has been changed and removed from conclusions either in the abstract and in the main text. However, an overview design can include in its structure a “suggestions for further research” section, for this reason, we find it very interesting for the reader and future research to summarise some key points to take into account when conducting research in ANS and MT field and overcome common shortcomings. 

Page 2 line 39, page 24 line 878

Intro

Again, please stick to the Authors' Guidelines. "(...) body’s internal environment (homeostasis) (1,2)." must read "(...) body’s internal environment (homeostasis) [1,2]." Revise thoroughly. Thank you for this clarification. Amended through all the manuscript

"Recent systematic reviews have shown that the MT approach is clinically effective (...)." and "iMoreover, several clinical studies have shown the effectiveness of MT (...)." Please see comments given above, and add information on the quality level of the referenced papers. Just "being published" is not considered a quality aspect.

 Thank you for this comment, text has been changed and adapted to the reviewer suggestion. 

Page 3 line 76

Same with "Evidence about autonomic effects of manual therapy interventions is synthesised within many reviews (41–46,48,49)." Again, provide detailed information, and revise carefully. Thank you for this clarification, Amended: references 41, 42,44 and 48 have been removed as they were rate as unclear RoB in our quality methodological assessment

page 4 line 125 

Methods

Did you note that your "Objectives" (aims) do not fully correspond to your Abstract section? Thank you for this clarification, corrections have been amended in the Abstract page 2 lines 20-21

regarding your inclusion criteria, please see comments given above. Revise carefully, and include aspects of utmost quality. We established eligibility criteria to ensure the inclusion of reviews conducted according to rigorous standards. We defined the characteristics of eligible systematic reviews according to specific characteristics discussed recently (DOI: 10.1186/s12874-019-0855-0).

We added a sentence at the ‘Potential biases …’ section (page 24 lines 858-860) to remark the appropriateness of our inclusion criteria.

"Cochrane Library, PubMed, EPISTEMONIKOS, and SCOPUS" would not seem exhaustive. Why didn't you include other databases? What about foreign languages? What about the grey literature? Recent data show that the best combination to identify SR in the context of overviews is to search MEDLINE and Epistmonikos complemented with reference checking (DOI: 10.1186/s12874-020-00983-3). 

We complemented this guidance with searches in scopus and WoS (the former, to track citations to relevant reviews).

We added a sentence at the ‘Potential biases …’ section (page 24 lines 854-858) to remark on the exhaustiveness of the search strategy.

"Two authors (SR and GA) independently assessed titles and abstracts of records identified by the electronic searches according to the inclusion criteria and decided on eligibility obtaining a full text copy from relevant references." Indeed, the two authors obviously simply searched. Again, what about the quality level of the included papers? We are compelled to disagree with this comment. At this step of evidence syntheses (SR, overviews and others) the search is completed, and researchers have to make decisions regarding study eligibility. It is recommended that at least two reviewers independently compare the reference yield by the search against inclusion criteria. We report this methodological standard according to the PRISMA statement (DOI: 10.1136/bmj.n71), as no specific reporting standards exist at the moment (the PRIOR statement is still in progress; DOI: 10.1186/s13643-019-1252-9)

"We solved disagreements involving a third author (FC) to reach a consensus through discussion." What kind of "disagreements" are you talking about? How often have there been "disagreements"? Was a consensus possible in all cases? And, again, what about the agreements regrading the quality levels? We have re-worded this sentence to be more specific. Quality issues do not apply at this step. We report this methodological standard according to the PRISMA statement (DOI: 10.1136/bmj.n71). Page 7 lines 228-230

Results

"All the reviews included randomised clinical trials (RCTs), one also included quasi-RCTs, four included non-randomised trials and two other designs." Again, see comments given above. "Quasi-RCTs, non-randomised trials and other designs" would not confirm Level-I research. Why did you include such papers? We provide a detailed rationale for this issue at the “General comment to “General remarks” from Rev#2”.

One of the functions of overviews is to provide a scope of the body of evidence that is included within the eligible reviews. Overview protocols have to establish inclusion criteria for the eligible systematic reviews but not for the primary research that they include. Overviews have to describe the characteristics of included reviews and critically synthesize their findings. We provided a detailed discussion on the limitations from original research included into the eligible reviews and provide a rationale for their implications.

With your Table 2 ("Characteristics of the included reviews"), please add your quality assessment. Again, please do not simply add the published papers; instead provide a sound, a valid, and a reasonable quality assessment, to allow to distinguish high- from poor-level papers. Amended: as suggested, table 2 has been modified and the RoB assessment of included SRs has been included. Page 43

Do not use Authors' names with your text, see "Schmid et al. found conflicting evidence (69)." Thank you for this clarification, Author’s names have been removed and replaced. 

There would seem some errors to be evident. See "and after mobilisations (41,45) and in the diastolic blood pressure after neck manipulations (Schmid et al. 2008)." Amended page 13 lines 460, 463

Same with "Galindez et al. found a decrease in systolic BP in hypertensive type I subjects but not in healthy participants (Galindez et al 2016) Two studies found an increase in HR after mobilisations (49,67), and one found no effect after cervical manipulation (Galindez et al 2017)." revise carefully. Amended page 13

 "Different quality assessment tools were used among the SRs (Appendix 4), showing a high heterogeneity of the studies' quality." See comments given above. First, the quality of the reviews would seem more important than the contents, to decide whether the paper will be included. Second, the results of your quality assessment must be provided here, and NOT somewhere in an Appendix. We have reformulated the table that describes the included reviews and the text to be clearer about this issue (both in the results and discussion sections). Again, the argument in this point is not if reviews should be included according to their bias or the bias of the primary research that they include. The overview has to make a clear picture of the reviews available to inform a clinical question and the scientific rigour behind them, and then discuss the implications for research and practice of such a body of evidence.

Table 4 page 49 (Risk of bias assessment of the included reviews (ROBIS)) + Sup 3 Table

Page 22 lines 773-777

From your Table 9 (!) it finally becomes clear that you ONLY found 1 (!) paper of "high quality". This would not seem convincing, not at all This comment is due to an (unwitting) misconception of domain-based approaches to assess the risk of bias. The judgements in the table refer to the risk of bias from included reviews, not to the quality of them.

We discussed this issue further above (choice of ROBIS instead of AMSTAR).

We have labelled “risk of bias of included reviews” the headings in the manuscript that could be a source of confusion.

On the other hand, a reformulated table 9 (now table 4) has been included to avoid semantic misconception of this construct. Page 49

Revise thoroughly for minor typos, see ""(...) are considered. (30,76,112)." Amended Page 21 line 754

 Again, do not use Authors' names, see "This tool is highly recommended when performing overviews or clinical guidelines (Whiting et al.). Amended Page 21 line 760

"The difference between those with low risk of bias and those with unclear risk of bias was due to the poor reporting of information, lack of protocol registration and methodological deficiencies." OK, but this would not seem an acceptable excuse. Good point to extend, some information regarding the methodological deficiencies of the SRs rated as UNCLEAR RoB has been added on page 21 line 771. 

"Our overview identified a high heterogeneity in regards to the quality of the articles included." Again, this would mean that papers with poor quality have been included. This must be elaborated more clearly. Moreover, it does not make sense to re-re-repeat those poor quality papers. With your revision, please stick exclusively to the Level-I papers. This refers to your conclusions. Papers with poor quality might be mentioned, but the drawbacks must be elucidated clearly. Please, see the comment below

"Consistent with the literature, the studies included in the present overview revealed the same issues, such as high heterogeneity, reporting deficiencies and the same articles rated differently." So please explain why you have included those papers? What about excluding such RCTs/systematic reviews? Remember that your write for the readers. What should a reader think after having swallowed your overview? Although we are aware that there are groups that encourage overview authors to include only high quality reviews, we also agree that this decision may introduce bias (doi: 10.1186/s13643-020-01509-0), specifically when the overview aims to scope the body of the evidence that informs a clinical question.

As we anticipated the topic of interest of our clinical question was assessed in several systematic reviews, we planned to conduct the overview to comprehensively identify them and then accurately appraise their bias and overlap. The process allowed us to map the existing reviews and the original research on this field and to develop a discussion on implications on the generability of available evidence.

 Please separate your "Suggestions" from the "Conclusions". Amended 

 Do not mix "Conclusions" and "Recommendations". All these aspects might be right, but do not mix it up, please.

 Good clarification, amended

Refs

- Stick exclusively to the Journal style, and revise for uniform formatting. Again and again, you state "[Internet]" and "Available from"; please delete. Consult some recently published Plos One papers. Provide doi and PMID numbers.

 We apologise for this inconvenience and references have been amended as suggested.

Concl

- With your Conclusions, please stick exclusively to your revised aims. Do not simply repeat your results here. Do not give a further literature review here. Instead, provide a reasonable and generalisable extension of your outcome.

 Information according to aims has been added to the conclusions section, Page 22 lines 805-810

Reviewer #3

I congratulate the authors for their work. The review addresses a relevant topic with important gaps in the literature. The compression of the mechanisms of effect of manual techniques is fundamental for the most adequate clinical reasoning and for the scientific advance in this subject. In fact, the relationship between autonomic effects and clinical improvement is an important gap in the literature. The large number of reviews on the subject can confuse readers and the authors knew how to unify the main findings of the studies in a very adequate way. However, I suggest better synthesising the ideas in the introduction and discussion to reduce the scope of the work. Below are other small suggestions

 We appreciate this comment from reviewer #3

Intro

1. Overall the introduction is very complete, but excessively detailed and long. Theoretically, readers interested in the topic should already have a basic knowledge of the topic, or they can consult the bibliographic references for further details. I suggest reducing paragraphs, or even unifying some (the second and third could unify action mechanisms - and gaps in these mechanisms - of manual therapy, without the need for so many explanations about the definition and clinical conditions that this area usually deals with - theme of the second paragraph ) We agree with the reviewer about the excessively detailed and long introduction. We understand that readers already interested in this topic may have basic knowledge on that, however, our intention when trying to publish in PLOS One journal was to broaden the audience to those that might be naïve to the topic. Is for this reason that we carefully considered introducing the topic for those who might find it hard to go through the references to have an idea of the importance of MT effects on ANS. 

Methods

2. The authors describe that they considered the level of evidence described in the reviews included, as well as the methodological quality of the studies included in the reviews, for the generalisation of evidence. However, the criteria used to summarise the level of evidence of the findings were not clear. I suggest defining a better instrument for this, something like the GRADE approach or similar. We agree that including judgements on the certainty of evidence would be informative, but there is no formal guidance to use GRADE for overviews. Besides, GRADE encourages the assessment of outcomes that have a direct impact on decision making and our clinical question could be, in part, out of its scope.

We have included a comment in the discussion related to this issue. Page 24 lines 869-865

On the other hand, we have clarified the efforts to appraise the body of evidence included in the eligible reviews by appraising their overlap and risk bias.

Results

3. Due to the amount of information (due to the large number of studies and variables), I suggest deleting the "newspaper" column from Table 2. Thank you, amended. Table 2 has been modified. page 43

4. Table 5. Review by Araujo et al, it seems that they would be "Intervertebral mobilisations" or "mobilisations" only, and not both. In this same table, some words in the techniques column are capitalised and others are not. Thank you for this clarification, amended. 

Discussion

5. As the introduction, the discussion is too long. Due to the number of studies and research questions, the results are already extensive. I suggest summarising the main topics both in the introduction and in the discussion. Many discussion topics have already been covered in the results. I suggest organising the discussion more broadly (without so many sections). I suggest organising the discussion according to the following topics: main findings; strenghts and limitations; comparison with previous studies; meaning of the study (clinical message and future directions).

 We understand the general feeling of being a long manuscript, however, an overview of SRs entails a deep assessment of all data available. This fact makes it difficult to synthesise and for this reason, the authors decided to structure the discussion into 4 different parts. In addition, to fully understand the autonomic effects it is necessary to distinguish each effect depending on the autonomic markers used and to discuss the appropriateness of each marker. The introduction section length has been justified in the first comment. Is for this reason that we would prefer to let both sections be like this. 

6. I see no reason for the authors to report results from more recently published RCTs….out of the scope of the study. We understand this comment that discussing results from recently published RCTs is out of the scope of the study. However, in the Overview design structure, there is a place to discuss agreements and disagreements with other studies or reviews. Also, we considered including useful information from very recent RCT’s in this field relevant. 

Journal Requirements

1. Please ensure that your manuscript meets PLOS ONE's style requirements, including those for file naming Done 

Thank you for stating the following financial disclosure: 

 [Funding for the publication of this article was provided by Registro de Osteopatas de España (ROE) www.osteopatas.org]. 

 Done 

We note that you have stated that you will provide repository information for your data at acceptance. Should your manuscript be accepted for publication, we will hold it until you provide the relevant accession numbers or DOIs necessary to access your data. If you wish to make changes to your Data Availability statement, please describe these changes in your cover letter and we will update your Data Availability statement to reflect the information you provide.

 Data availability will be upon request

Please include a separate caption for each figure in your manuscript.

 Amended

 Please include captions for your Supporting Information files at the end of your manuscript, and update any in-text citations to match accordingly. 

 Amended

---

## [Decision Letter · Decision Letter 1]

1 Nov 2021

PONE-D-21-24377R1Do manual therapies have a specific autonomic effect? An overview of systematic reviews.PLOS ONE

Dear Dr. Roura,

Thank you for submitting your manuscript to PLOS ONE. After careful consideration, we feel that it has merit but does not fully meet PLOS ONE’s publication criteria as it currently stands. Therefore, we invite you to submit a revised version of the manuscript that addresses the points raised during the review process.

Having intensively reviewed your revised draft, our external reviewers agreed with their final recommendations. Additionally, I have double checked your revised version, to come to a final decision (see R #1). All in all, I am convinced that your revised paper will be worth following, even if your revised version still would benefit from minor and major re-edits and some polishing.

We look forward to receiving your revised manuscript.

Kind regards,

Andrej M Kielbassa

Academic Editor

PLOS ONE

Reviewers' comments:

Reviewer's Responses to Questions

**Comments to the Author**

1. If the authors have adequately addressed your comments raised in a previous round of review and you feel that this manuscript is now acceptable for publication, you may indicate that here to bypass the “Comments to the Author” section, enter your conflict of interest statement in the “Confidential to Editor” section, and submit your "Accept" recommendation.

Reviewer #1: All comments have been addressed

Reviewer #2: (No Response)

Reviewer #3: All comments have been addressed

2. Is the manuscript technically sound, and do the data support the conclusions?

Reviewer #1: Yes

Reviewer #2: No

Reviewer #3: Yes

3. Has the statistical analysis been performed appropriately and rigorously? 

Reviewer #1: Yes

Reviewer #2: N/A

Reviewer #3: Yes

4. Have the authors made all data underlying the findings in their manuscript fully available?

Reviewer #1: Yes

Reviewer #2: Yes

Reviewer #3: Yes

5. Is the manuscript presented in an intelligible fashion and written in standard English?

Reviewer #1: Yes

Reviewer #2: Yes

Reviewer #3: Yes

6. Review Comments to the Author

Reviewer #1: All comments have been addressed by the authors making the paper acceptable for publication in its current form

Reviewer #2: General remark

- This revised and re-subitted draft has considerably improved.

- Moreover, the Authors have thoroughly responded to this reviewer's previous comments. Their thoughts would seem comprehensible, even if some few positions are considered debatable. The future readers will decide on those aspects.

- Unfortunately, still some aspects are in need of revision. Please see below, and remember that this is not the right place to persistently insist on prevailing your ideas.

Abstract

- Again, please stick to Journal guidelines. Maximum word count is 300 (but NOT 399!). Please revise carefully. Remember that your answer ("Thank you for this comment, Abstract has been shortened considerably") would not seem satisfying.

- Again, please adapt aims given with the Abstract section and the questions asked in the Introduction section. This reviewer has read your comments ("Thank you for this clarification, corrections have been

amended in the Abstract page 2 lines 20-21"), but still there would be some need to polish this topic.

- Remember that with your conclusions, answers must be given exclusively to your questions.

Introduction

- Please see comments given above.

Materials and Methods

- Heading must read "Materials and methods". Again, please stick to the Journal guidelines, and consult some recently published Plos One papers. "METHODS" would not seem acceptable. Remember that there will not be any thorough copy editing, so only flawless papers will be acceptable.

- As a general recommendation, please stick to Journal style with ALL other aspects. For example, you repeatedly refer to "table 2", "table 3", "table 4", and so on. Again, please consult some recently published Plos One papers, there you will see that this must read "Table (1, 2, 3, or 4)". Same with other minor aspects, please revise carefully, and remember that submitting a flawless manuscript is considered the Authors' task, and that the typesetter will not be able to copy edit your draft.

Conclusions

- Do not give a "summary" here. For example, "There is considerable research (...)" and "This overview summarized the information reported by 12 systematic reviews." are not considered conclusions.

- Again, with your Conclusions section, please stick exclusively to your revised aims. Do not simply repeat your results here. Instead, provide a reasonable and generalizable extension of your outcome.

- Same thoughts are valid for "Implications for research: (...)". These aspects surely would seem right, and should be transferable to the Discussion section. However (and again), those thoughts are not considered Conclusions. Again, please revise carefully.

References

- Again, revise for uniform formatting. Stick to Journal style. Provide doi and PMID numbers. Again, the Authors' response ("We apologise for this inconvenience and references have been amended as suggested.") would not seem satisfying.

- Example would be: "Cheng L, Weir MD, Xu HH, Antonucci JM, Lin NJ, Lin-Gibson S, et al. Effect of amorphous calcium phosphate and silver nanocomposites on dental plaque microcosm biofilms. J Biomed Mater Res B Appl Biomater. 2012; 100(5): 1378–1386. https://doi.org/10.1002/jbm.b.32709 PMID: 22566464" Revise thoroughly, but carefully. Remember that it is not the idea to re-re-review your submitted draft, and another re-submission not considered satisfying would lead to "outright rejection".

In total, this revised and resubmitted draft still is not considered ready to proceed. The Authors should be given another try to re-submit a perfect manuscript considered acceptable,

Reviewer #3: I congratulate the authors for the revised version.

I maintain my comments on the length of the introduction and discussion. I believe the text could be better synthesized. However, I agree with the authors' arguments.

7. PLOS authors have the option to publish the peer review history of their article (what does this mean?). If published, this will include your full peer review and any attached files.

Reviewer #1: No

Reviewer #2: No

Reviewer #3: No

---

## [Author Response · Author response to Decision Letter 1]

8 Nov 2021

Comments Responses

Rev#2

General remark 

Response: 

Thank you for giving us the opportunity to keep improving the manuscript

Abstract

Again, please stick to Journal guidelines. Maximum word count is 300 (but NOT 399!). Please revise carefully. Remember that your answer ("Thank you for this comment, Abstract has been shortened considerably") would not seem satisfying 

response: Amended. The abstract counts 267 words.

Again, please adapt aims given with the Abstract section and the questions asked in the Introduction section. This reviewer has read your comments ("Thank you for this clarification, corrections have beenamended in the Abstract page 2 lines 20-21"), but still there would be some need to polish this topic. 

Response: Amended.

Remember that with your conclusions, answers must be given exclusively to your questions 

Response: Amended

Introduction

Please see comments given above. Amended Page 4 lines 129-133

Materials and Methods

Heading must read "Materials and methods". Again, please stick to the Journal guidelines, and consult some recently published Plos One papers. "METHODS" would not seem acceptable. Remember that there will not be any thorough copy editing, so only flawless papers will be acceptable 

Response: 

Amended Page 6 line 154

As a general recommendation, please stick to Journal style with ALL other aspects. For example, you repeatedly refer to "table 2", "table 3", "table 4", and so on. Again, please consult some recently published Plos One papers, there you will see that this must read "Table (1, 2, 3, or 4)". Same with other minor aspects, please revise carefully, and remember that submitting a flawless manuscript is considered the Authors' task, and that the typesetter will not be able to copy edit your draft. 

Response: 

Tables, figures and supplementary information has been adapted to the journal requirements from https://journals.plos.org/plosone/s/tables

https://journals.plos.org/plosone/s/supporting-information

https://journals.plos.org/plosone/s/submission-guidelines

See highlighted marks. 

Conclusions

Do not give a "summary" here. For example, "There is considerable research (...)" and "This overview summarized the information reported by 12 systematic reviews." are not considered conclusions.

Response:

Conclusions section has been amended. Page 25

Again, with your Conclusions section, please stick exclusively to your revised aims. Do not simply repeat your results here. Instead, provide a reasonable and generalizable extension of your outcome. 

Response: 

Conclusions section has been amended. Page 25

Same thoughts are valid for "Implications for research: (...)". These aspects surely would seem right, and should be transferable to the Discussion section. However (and again), those thoughts are not considered Conclusions. Again, please revise carefully. 

Response: 

Amended. Mentions to Implications for research have been moved to Discussion.

References

- Again, revise for uniform formatting. Stick to Journal style. Provide doi and PMID numbers. Again, the Authors' response ("We apologise for this inconvenience and references have been amended as suggested.") would not seem satisfying

- Example would be: "Cheng L, Weir MD, Xu HH, Antonucci JM, Lin NJ, Lin-Gibson S, et al. Effect of amorphous calcium phosphate and silver nanocomposites on dental plaque microcosm biofilms. J Biomed Mater Res B Appl Biomater. 2012; 100(5): 1378–1386. https://doi.org/10.1002/jbm.b.32709 PMID: 22566464" Revise thoroughly, but carefully. Remember that it is not the idea to re-re-review your submitted draft, and another re-submission not considered satisfying would lead to "outright rejection 

Response: 

Amended.

---

## [Decision Letter · Decision Letter 2]

15 Nov 2021

Do manual therapies have a specific autonomic effect? An overview of systematic reviews.

PONE-D-21-24377R2

Dear Dr. Roura,

We’re pleased to inform you that your manuscript has been judged scientifically suitable for publication and will be formally accepted for publication once it meets all outstanding technical requirements.

Kind regards, congratulations and compliments, and stay healthy

Andrej M Kielbassa, Prof. Dr. med. dent. Dr. h. c.

Academic Editor

PLOS ONE

Additional Editor Comments (optional):

Reviewers' comments:

Reviewer's Responses to Questions

**Comments to the Author**

1. If the authors have adequately addressed your comments raised in a previous round of review and you feel that this manuscript is now acceptable for publication, you may indicate that here to bypass the “Comments to the Author” section, enter your conflict of interest statement in the “Confidential to Editor” section, and submit your "Accept" recommendation.

Reviewer #2: All comments have been addressed

2. Is the manuscript technically sound, and do the data support the conclusions?

Reviewer #2: Yes

3. Has the statistical analysis been performed appropriately and rigorously? 

Reviewer #2: Yes

4. Have the authors made all data underlying the findings in their manuscript fully available?

Reviewer #2: Yes

5. Is the manuscript presented in an intelligible fashion and written in standard English?

Reviewer #2: Yes

6. Review Comments to the Author

Reviewer #2: The Co-Authors have satisfyingly revised their draft, according to the previous comments. All other reviewers have agreed to accept this submission, which is considered ready to proceed now.

7. PLOS authors have the option to publish the peer review history of their article (what does this mean?). If published, this will include your full peer review and any attached files.

Reviewer #2: No

---

## [Editor Report · Acceptance letter]

17 Nov 2021

PONE-D-21-24377R2 

Do manual therapies have a specific autonomic effect? An overview of systematic reviews 

Dear Dr. Roura:

I'm pleased to inform you that your manuscript has been deemed suitable for publication in PLOS ONE. Congratulations! Your manuscript is now with our production department. 

Kind regards, 

on behalf of

Prof. Dr. med. dent. Dr. h. c. Andrej M Kielbassa 

Academic Editor

PLOS ONE